

# Aerosol Mass yields of selected Biogenic Volatile Organic Compounds – a theoretical study with near explicit gas-phase chemistry

Carlton Xavier[1], Anton Rusanen[1], Putian Zhou[1], Chen Dean[1], Lukas Pichelstofer[1], Pontus Roldin[2], Michael Boy[1]

[1]Institute for Atmospheric and Earth Systems Research (INAR), Physics, University of Helsinki

[2]Division of Nuclear Physics, Lund University, Box 118, SE-22100, Lund, Sweden

**Correspondence** : Carlton Xavier (carlton.xavier@helsinki.fi), Michael Boy (michael.boy@helsinki.fi)

## Abstract

In this study we modeled secondary organic aerosols (SOA) mass loadings from the oxidation (by $O_3$, OH and $NO_3$) of five representative Biogenic Volatile Organic compounds (BVOCs): isoprene, endocyclic bond containing monoterpenes (α-pinene and limonene), exocyclic double bond compound (β-pinene) and a sesquiterpene (β-caryophyllene). The simulations were designed to replicate idealized smog chamber and oxidative flow reactors (OFR). The master chemical mechanism (MCM) together with the peroxy radical autoxidation mechanism (PRAM), were used to simulate the gas-phase chemistry. The aim of this study was to compare the potency of MCM and MCM+PRAM in predicting SOA formation. SOA yields were in good agreement with experimental values for chamber simulations when MCM+PRAM was applied, while a standalone MCM under-predicted the SOA yields. Compared to experimental yields, the OFR simulations using MCM+PRAM over-predicted SOA mass yields for BVOCs oxidized by $O_3$ and OH, probably owing to increased seed particle surface area used in the OFR simulations. SOA yields increased with decreasing temperatures and NO concentrations and vice-versa. This highlights the limitations posed when using fixed SOA yields in a majority of global and regional models. Few compounds that play a crucial role (>95% of mass load) in contributing to SOA mass increase (using MCM+PRAM) are identified. The results further emphasized that incorporating PRAM in conjunction with MCM does improve SOA mass yields estimation.

## 1. Introduction

Atmospheric secondary organic aerosols, formed from gas to particle phase conversion of the oxidation products of volatile organic compounds (VOC) significantly impact the organic aerosol mass loadings (Griffin, 1999; Kanakidou et al., 2005). However, the scale of SOA contribution to the aerosol particle mass is still subjected to high uncertainties (Hao et al., 2011). The elevated aerosol particle





concentrations are shown to have inimical effects on health (Miller et al., 2007), and a varying degree of

influence on the climate by forming cloud condensation nuclei (CCN), altering the cloud properties and

radiative balance (Rosenfeld et al., 2014; Schmale et al., 2018). Therefore, it is acutely necessary to

understand the role and contributions of SOA to the particle loading in the atmosphere. Biogenic VOCs from

forest are estimated to contribute to about 90% of VOCs emissions globally (Guenther et al., 1995, 1999 and

2000). The most important BVOCs for SOA formation are isoprene ($C_5H_8$), monoterpenes ($C_{10}H_{16}$) and

sesquiterpenes ($C_{15}H_{24}$). These compounds are all alkenes containing at least one carbon-carbon double bond,

enabling them to undergo oxidation by the dominant atmospheric oxidants: the hydroxyl radical (OH), ozone

($O_3$) and the nitrate radical ($NO_3$). For some of the terpenes, initial oxidation steps can lead to formation of

highly oxygenated organic molecules (HOM). These HOMs generally have low volatilities and can condense

nearly irreversibly, thereby producing SOA (Ehn et al., 2014). HOMs, detected in both the ambient

atmosphere and chamber experiments (Ehn M, et al., 2012) are formed by autoxidation (Berndt et al., 2016;

Crounse and Nielsen, 2013) wherein peroxy radicals ($RO_2$) undergo subsequent intramolecular H-shifts

accompanied by rapid reactions with $O_2$. Autoxidation hence results in compounds containing multiple

functional groups such as hydroxyls, peroxides and carbonyls (Bianchi et al., 2017,Bianchi et al., 2019).

A majority of chamber and flow-tube experiments have focused on HOM formation from the

oxidation of various VOCs and their contribution to SOA mass loadings (Ehn et al., 2014; Kristensen et al.,

2017). Oxidation of isoprene (Liu et al., 2016), endocyclic monoterpenes containing reactive double bonds

such as α-pinene and limonene (Zhao et al., 2015) , or exocyclic double bond containing compounds such as

β-pinene (Jokinen et al., 2015) and sesquiterpenes such as β-caryophyllene (Chen et al., 2012) have been

investigated. The SOA forming potential of various BVOCs depends on the isomeric structures (Friedman

and Farmer, 2018; Keywood et al., 2004). Ozonolysis of compounds containing reactive endocyclic bonds

such as α-pinene produce higher SOA mass yields of 41% in comparison to those with exocyclic bonds (β-

pinene), which produce mass yields of 17 % (Lee et al., 2006). One explanation for this dependence on the

isomeric structure is attributed to the formation of HOMs (Ehn et al., 2014). Another important factor

influencing HOM formation is the initial oxidant, as pointed out by Zhao and co-workers (2015). They

showed that the SOA formation by OH oxidation  of α-pinene and limonene were lower when compared to

their SOA formed by ozonolysis. Further they measured lower H/C ratio for SOA produced by monoterpene

ozonolysis (experiments were carried out in dark with CO as OH scavenger), in comparison to OH oxidation

of α-pinene and limonene. This was attributed to the formation of $RO_2$ radicals (monoterpenes +$O_3$) which

undergo internal hydrogen shifts and subsequently react with another $RO_2$ radical, to form compounds

containing carbonyl groups while losing hydrogen atoms in the process. A similar analysis was conducted by



Draper et al. (2015), who showed that an increase in $NO_2$ concentration reduced α-pinene ozonolysis SOA
mass yields, while no appreciable reduction in mass yields are reported for β-pinene and $\Delta^3$- carene
ozonolysis. On the other hand, the mass yields from limonene ozonolysis increased with increasing $NO_2$
concentrations (Draper et al., 2015). This disparity in mass yields for different BVOCs in the presence of $NO_2$
is possibly caused by the formation of high MW oligomers (or lack of in case of α-pinene) through oxidation
with $NO_3$ that contribute to SOA mass loadings (Draper et al., 2015).
Due to computational limitations, many regional and canopy scale atmospheric chemistry models
generally use isoprene and/or a representative monoterpene (generally α-pinene), to model SOA yields
(Friedman and Farmer, 2018). The SOA yields of different monoterpenes vary with structure, $NO_x$ and
temperature (Friedman and Farmer, 2018; Kristensen et al., 2017; Presto et al., 2005). This poses a limitation
on using representative monoterpene fixed SOA yields in many of the global models and increases
uncertainties in predicting cloud condensation nuclei concentrations, cloud droplet number concentrations and
radiative balance due to aerosol loading's.
This work aims to investigate the SOA mass loading from the oxidation products of BVOCs with the
atmospheric oxidants OH, $O_3$ and $NO_3$ with a specific focus on the BVOCs isoprene, α-pinene, β-pinene,
limonene and β-caryophyllene. Further we study the effect of varying temperature (258.15 K – 313.15 K) and
NO concentrations (0 - 5 ppb) on α-pinene oxidation mass yields. We use the master chemical mechanism
(MCMv3.3.1) (Jenkin et al., 1997, 2012 and 2015; Saunders et al., 2003), a near explicit gas-phase chemical
mechanism together with peroxy radical autoxidation mechanism (PRAM, Roldin et al., 2018) (PRAM +
MCM). The aim is to understand the importance and contribution of peroxy radical autoxidation products to
the SOA mass yields from terpenes.

2. **Model description**
2.1 **Malte Box**
MALTE (Model to predict new Aerosol formation in Lower TropospherE) is a one-dimensional
model consisting of modules calculating boundary layer meteorology, emissions of BVOCs, gas-phase
chemistry and aerosol dynamics with the aim to simulate particle distribution and growth in the lower
troposphere (Boy et al., 2006). In this study, a zero-dimensional version, MALTE-Box is applied to simulate
an ideal chamber and flow-tube environment (i.e. no wall losses effects are considered in this study). For the





simulations performed in this study the emission module was switched off while only employing the gas-
phase chemistry and aerosol dynamics module.

Kinetic preprocessor (KPP) is used to generate a system of coupled differential equations to solve the

gas-phase chemistry schemes (MCM+PRAM, Damian et al., 2002). The peroxy radical autoxidation
mechanism (PRAM), (Roldin et al., 2018), formulated based on the oxidation of monoterpenes as described
by Ehn et al. (2014) was incorporated alongside MCMv3.3.1. PRAM describes the evolution of peroxy
radicals ($RO_2$) from the ozonolysis of monoterpenes driven by subsequent H-shifts and $O_2$ additions. The
current version of PRAM considers HOM autoxidation for a fraction of the peroxy radicals formed during the
ozonolysis of α-pinene and limonene and OH oxidation of α-pinene, β-pinene and limonene. PRAM considers
temperature dependent autoxidation reaction rates, which is important when investigating the SOA mass
yields at varying temperatures (Table 1c). It should be noted that the temperature dependence in PRAM is a
first of its kind but needs further evaluation using recent measurements of HOM formation at different
temperatures (e.g. Quéléver et al.2018).

The aerosol dynamics are simulated using the University of Helsinki Multicomponent Aerosol model

(UHMA) originally from Korhonen et al. (2004). The model has undergone significant development since
then to allow simulation with all the compounds from MCM. It now supports an unlimited number of
condensing vapors and solves condensation using the analytical predictor of condensation method from
Jacobson (1997). The condensation algorithm considers both, the Kelvin effect and Raoult's law. The
processes included in the model are nucleation, condensation, evaporation, coagulation and deposition. The
discretization of the size distribution and the time evolution is modeled with the moving section approach,
with optional redistribution to a fixed grid. In this work, the redistribution is active to make the coagulation
more accurate, since it requires that grid points are available near the size of the coagulated particles. In this
study nucleation and deposition are not active, and hence are not considered. A total of 100 size bins ranging
from 1nm to 20μm with the fixed grid was applied for this study.

A group contribution method based on Nannoolal et al. (2008) using the UManSysProp online system

(Topping, 2016) was used to estimate the pure liquid saturation vapor pressures ($p_0$) of the organic
compounds in MCMv3.3.1. For the PRAM species, $p_0$ were estimated using the functional group method
SIMPOL (Pankow and Asher, 2008; see Roldin et al., 2018 for details)
2.2 **Simulations**

The simulations performed in this study are aimed to closely resemble an idealized smog chamber

(batch mode setup) and an Oxidative Flow Reactor (OFR) without interactions between the gas phase and the



system walls. For the chamber runs, the VOC and oxidants were introduced at the beginning (time, t=0 sec),
set to certain concentrations and then allowed to react. Both chamber and OFR simulations are performed
using ammonium sulfate seed particles which are introduced at time t=0. The condensation sink (CS) was
inferred from the size distribution of seed particles used in the model. The CS for the chamber and OFR
simulations was set to 0.00067 s$^{-1}$ and 0.067 s$^{-1}$ respectively.  SOA mass yields obtained using an OFR are
sensitive to short residence time used, hence the seed particle surface area should be chosen in order to
overcome the mass yield underestimation (Ahlberg et al., 2019). CS sensitivity runs (Supplement Figure S1)
were performed for α-pinene-$O_3$ to determine the CS for which there are no appreciable change in mass yields
with increasing particle surface.
The simulation for the chamber setup is run for a maximum time of 24 hours and ends when either of
the 2 criteria are satisfied: (1) the simulation time reaches the 24-hour mark or (2) 90 % of the initial
precursor VOC has reacted away. In the latter case the simulation is continued for an additional 2 hours to
ensure enough time for the vapors to condense onto the seed particles. On the contrary, the OFR runs were
simulated for a maximum residence time of 100 seconds, ensuring all initial precursor vapors were oxidized.
Seed particles were also added in the OFR simulations. The oxidant concentrations used for the OFR
simulations are significantly higher in comparison to the simulated chamber runs (~2 orders of magnitude
larger). The time step for the chamber and flow-tube simulations are set to t=10 s and t =0.1 s respectively.
The runs performed were oxidant specific (i.e. VOCs would be oxidized by only one specific oxidant at any
given time). For the $O_3$ specific simulations no OH could form in both, OFR and chamber setups, thus
enabling oxidation of $O_3$ to be the only pathway.
The simulations were performed at atmospheric relevant $NO_x$ ($NO_x$ = NO +$NO_2$ ) concentrations,
corresponding to [NO]=0.5 ppb and [$NO_2$] = 2.0 ppb conditions with the relative humidity (RH) set to 60 %
and temperature to 293.15 K. The RH value considered in this study is based on previous published
experimental studies performed at ~60 % in both smog chamber (Bruns et al., 2015; Ehn et al., 2014;
Stirnweis et al., 2017) and OFR (Ahlberg et al., 2019). α-pinene ozonolysis runs were performed at four
different temperatures: 258.15 K, 278.15 K, 303.15K and 313.15 K, respectively. SOA mass yields are
expected to increase with decreasing temperature (Saathoff and Naumann, 2009). A similar temperature
dependence was observed by Kristensen et al. (2017) who observed SOA mass yield from α-pinene
ozonolysis at ~ 40 % and ~20 % at 258 K and 293 K respectively. Analogous to analyzing the effect of
varying temperature on SOA yields, we study the variation in α-pinene ozonolysis SOA mass yields by
varying the $NO_x$ concentrations. SOA yields for α-pinene ozonolysis at high $NO_x$ conditions should be
suppressed (Ng and Chhabra, 2007), which could be due to the production of relatively, volatile organic





nitrates under high NO$_x$ conditions as compared to less volatile products during low NO$_x$ conditions (Presto et
al., 2005).
Furthermore, two different chemistry schemes were applied for the simulations. One scheme consisted
of only the MCM chemistry mechanism and the second included the MCM+PRAM chemistry mechanism.
Table 1a shows the concentrations of different BVOCs and Table 1b shows the oxidants concentrations used
for the simulations.
**Table 1a**. Concentrations of different BVOCs

| α-pinene (ppb) | β-pinene (ppb) | Isoprene (ppb) | Limonene (ppb) | β-caryophyllene (ppb) |
|---|---|---|---|---|
| 0.5, 1.0, 5.0, 50.0, 100.0, 200.0 | 0.5, 1.0, 5.0, 50.0, 100.0, 200.0 | 5.0, 50.0, 100.0, 200.0 | 1.0, 5.0, 50.0, 100.0, 200.0 | 0.5, 1.0, 2.0, 5.0, 10.0 |


**Table 1b**. Concentrations of different oxidants for chamber and flow-tube runs

| OH (* 10$^6$ #/cm$^3$) - chamber<br>OH (* 10$^8$ #/cm$^3$) - OFR | O$_3$ (* 10$^{11}$ #/cm$^3$) - chamber<br>O$_3$ (* 10$^{13}$ #/cm$^3$) - OFR | NO$_3$ (* 10$^7$ #/cm$^3$) - chamber<br>NO$_3$ (* 10$^9$ #/cm$^3$) - OFR |
|---|---|---|
| 2.0, 5.0 ,10.0, 50.0,100.0 | 1.0, 5.0 ,10.0, 50.0,100.0 | 1.0, 5.0 ,10.0, 50.0,100.0 |


**Table 1c**. NO concentrations and temperatures used for α-pinene ozonolysis

| NO (ppb) | 0.5 (default), 0, 0.2, 1, 2, 5 |
|---|---|
| Temperature (K) | 293.15 (default), 258.15, 278.15, 303.15, 313.15 |


2.3 **Mass Yields**
The SOA mass yields (Y) are determined by calculating the ratio of the amount of SOA or mass
concentration of organic aerosol formed (C$_{OA}$) to the amount of VOC (ΔVOC) reacted:
$$Y = \frac{C_{OA}}{\Delta VOC} \qquad (1)$$



A volatility basis set is fit to the data to obtain the volatility distribution. In this study equilibrium
partitioning was only assumed for deriving the volatility distribution based on the model simulations.
Following Donahue et al. (2006), the SOA is assumed to be in equilibrium with the gas-phase and using the
effective saturation concentration $C_i^*$ spaced logarithmically. The individual product partitioning to the
particle phase can be estimated using
$$E_i = \left(1 + \frac{C_i^*}{C_{OA}}\right)^{-1} \qquad (2)$$

Where $E_i$ is the fraction of species in the condensed particle phase. The above equation determines the
fraction of species in the particle phase as well as in the gas phase. For example, if we assume $C_{OA}$ =10 µg m$^{-3}$
a species with $C^*$ = 10 µg m$^{-3}$ will partition 50 % to condensed phase and the rest 50% will reside in the gas
phase. The fidelity of this equilibrium partitioning enables the parameterization of product vapors in volatility
$C^*$ bins that are near the $C_{OA}$ concentrations (Henry et al., 2012).

**3. Results and Discussion**
**3.1 BVOCs – O$_3$ chamber and flow-tube simulations**
SOA mass yields were simulated for the oxidation of various biogenic volatile organic compounds
(isoprene, α-pinene, limonene and β-caryophyllene, β-pinene) by dominant atmospheric oxidants OH, O$_3$ and
NO$_3$. The following section examines the comparison between the yields derived using MCM+PRAM and a
standalone MCM for chamber and flow-tube experiments. In Fig 1. the upper panel indicates MCM+PRAM
contribution to SOA mass from BVOCs ozonolysis and the lower panel shows the differences between the
yields obtained from the MCM+PRAM (Y$_{MCM+PRAM}$) and MCM scheme (Y$_{MCM+PRAM}$ − Y$_{MCM}$).
The abscissa, depicted on a log scale, considers the entire range of SOA mass loadings from 1-1150
µg/m$^{-3}$. Each data point is representative of simulated SOA mass yields resulting from variable BVOC
loading. The resulting mass yields for α-pinene in the range shown in Table 2a are consistent with the yields
found in various smog chamber experiments. Kristensen et al. (2017) found mass yields of 0.22 and 0.21 for
α-pinene ozonolysis at 293 K with SOA mass loadings of 62 and 59 µg m$^{-3}$, respectively, while Pathak et al.
(2007) found similar mass yields values in the range of 0.16 – 0.21 for mass loadings between 33.7 – 50.7 µg
m$^{-3}$. These values are within the range of mass yields derived using MCM+PRAM (0.15 - 0.2) while those
derived with only MCM severely under-predict the mass yields. The MCM+PRAM also shows better
agreement with experiments when estimating the lower range mass yields for SOA mass loadings of < 15 µg





m$^{-3}$. This is supported by the values obtained by Shilling et al. (2008), where the authors measured a 0.09
yield from α-pinene ozonolysis for SOA mass loading of 10.6 µg m$^{-3}$. Limonene ozonolysis mass yields using
MCM+PRAM in comparison to standalone MCM, are much closer to the values given by Waring (2016),
wherein he measured high yields of 0.26 - 1.06 for SOA loadings of 1.7 - 718.4 µg m$^{-3}$.
The formation of HOM from β-pinene ozonolysis is low (Ehn et al., 2014; Jokinen et al., 2015) and
hence not considered in PRAM. The peroxy radical autoxidation mechanism for β-caryophyllene ozonolysis
has not yet been developed and hence is not considered in PRAM. Chen et al. (2012) measured SOA particle
mass yield for β-caryophyllene ozonolysis in the range of 0.1- 0.2 for mass loadings $M_{org}$ < 10 µg m$^{-3}$ and
ascertained the $O_3$ concentration insensitivity to the yields. Pathak et al. (2008) measured the β-pinene
ozonolysis mass yields in the range of 0.013- 0.081 for lower reacted concentrations (8-40 ppb) of β-pinene,
while Griffin (1999) measured a yield from 0.01-0.17 for reacted β-pinene in the range of 30 -180 ppb. When
comparing the measured mass yield values for β-caryophyllene and β-pinene ozonolysis to the modeled
values using the MCM scheme, it is evident that the MCM scheme drastically under-predicts the SOA mass
yields.
Today oxidation flow reactor (OFR) experiments are complementing the traditional batch mode smog
chamber experiments. The OFR generally exhibits lower mass yields compared to the smog chamber
experiments at ranges of equivalent oxidant exposure (Lambe et al., 2015). We modeled flow-tube simulation
after the potential aerosol mass (PAM) OFR, where the residence time is in the order of a few to several
minutes (Lambe et al., 2011). Our model simulations are performed with a maximum residence time of 100
seconds with $O_3$ exposures ranging from $1.0 \times 10^{15} - 1.0 \times 10^{17}$ molecules cm$^{-3}$ s (residence time x [$O_3$]).
Kang and Root (2007) measured a value of 0.2 for ozonolysis of α-pinene for an initial precursor VOC
concentration of 100 ppbv, while we obtain 0.25-0.3 for the similar initial precursor concentrations. The OFR
yields for β-pinene are significantly lower (0.02) than the values measured by Kang and Root (2007) wherein
they measured a yield of 0.49 for similar initial precursor concentrations. Addition of seed particles promotes
condensation, leading to increased SOA yields (Lambe et al., 2015) which was confirmed by Ahlberg et al,
(2019). Kang and Root (2007) found that using seed particles, the yield from α-pinene ozonolysis increased
by a factor of ~1.4 which can explain our yields for α-pinene ozonolysis simulations, while the absence of a
PRAM for β-pinene could explain the low values from our simulation. The mass spectra plot (Figure S2)
shows that PRAM contributes the majority of dimers to the particle phase, while MCM dominate monomer
contribution. Another interesting facet of Figure S2 are the different condensing compounds in both OFR and
chamber simulations. The higher absolute $RO_2$ concentrations in the OFR simulations explain the lower
concentration of HOM monomers and dimers relative to the chamber simulations, i.e. the high $RO_2$



concentrations in the OFR cause termination of the peroxy radical autoxidation chain before the $RO_2$ become highly oxygenated, thereby influencing SOA yields. Hence, this should be taken into account when using yields from OFR as inputs to regional and global models.

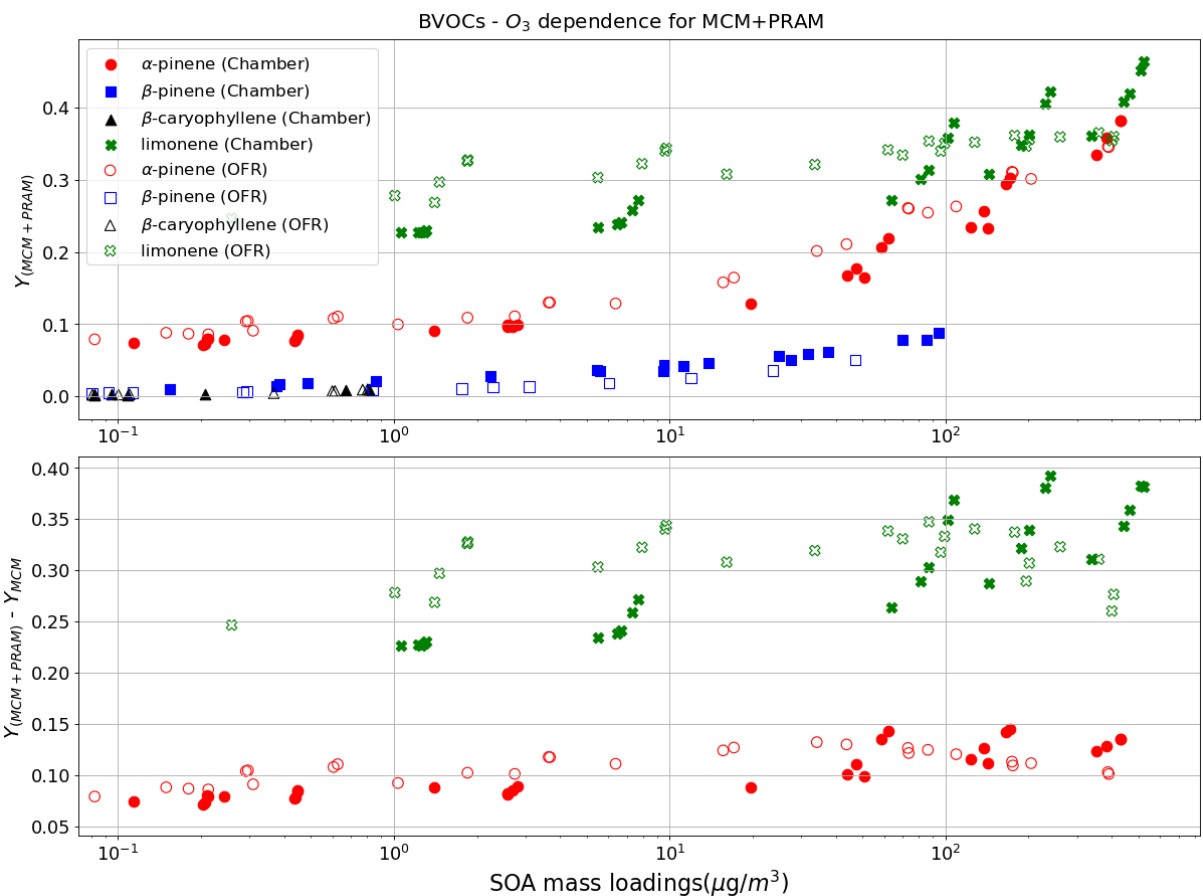

**Figure 1.** The mass yields from the ozonolysis of BVOCs modelled after chamber and flow-tube settings. The figure shows a comparison of SOA mass yields obtained from simulations with MCM + PRAM and PRAM. The BVOCs are represented by α-pinene (red circles), β-pinene (blue squares), limonene (green stars) and β-caryophyllene (black triangles) with the filled and open symbols depicting the chamber and oxidation flow reactor (OFR) simulations respectively.

**Table 2.** Mass yields for BVOCs ozonolysis at 293 K for different range of mass loadings using a chamber[†] and OFR[∥] setup.

| SOA mass loading (μg m⁻³) | MCM + PRAM mass yields range | MCM mass yields range | BVOC | Experimental yields | References |
|---|---|---|---|---|---|





| 0 - 15[†] | 0.07– 0 .08 | 0.00 – 0.06 | α-pinene | 0.09 (10.6) | Shilling et al. (2008) |
|---|---|---|---|---|---|
| 16 - 60[†] | 0.12 – 0.20 | 0.06 – 0.11 | α-pinene | 0.16 - 0.21 | Pathak et al. (2007) |
| 61 – 200[†] | 0.22 – 0.30 | 0.12 – 0.15 | α-pinene | 0.22 (62) | Kristensen et al. (2017) |
| 1.1– 550[†] | 0.24 -0.48 | 0.007-0.06 | limonene | 0.26-1.06 | Waring (2016) |
| 0-100[‖] | 0.07-0.25 | 0-0.13 | α-pinene | 0.2 | Kang and Root (2007) |

### 3.2 BVOCs – OH chamber and flow-tube simulations

The $\Delta Y$ in the lower panel of Figure 2 for limonene, β-pinene and α-pinene resulting from the fact that PRAM considers peroxy radical autoxidation product formation from OH oxidation of these BVOC's. The results for SOA mass yields of OH oxidation of α-pinene using MCM+PRAM is close to the measured values (Kristensen et al., 2017), while using only MCM under-predicts the mass yields. Similarly, the current lack of peroxy radical autoxidation product mechanism for β-caryophyllene and isoprene result in $\Delta Y = 0$ values for PRAM. The maximum SOA mass yield for OH oxidation of α-pinene is lower than the yield from ozonolysis which is suspected to arise due to the formation of more volatile oxidation products produced during OH oxidation (Bonn and Moortgat, 2002; Kristensen et al., 2014). For β-caryophyllene, Griffin (1999) measured SOA mass yields in the range of 0.37 – 0.79 for mass loadings of 17 - 82 $\mu g m^{-3}$ while Tasoglou and Pandis (2015) measured yields around 0.2 for photo-oxidation at low $NO_x$ conditions for loadings of 10 $\mu g m^{-3}$. These values are in good agreement with our results which show a similar yield range of 0.3 – 0.7 for loadings between 20 - 100 $\mu g\ m^{-3}$ and 0.21 for loading of 10 $\mu g\ m^{-3}$. Currently there are no experiments providing HOM yields from experiments of OH oxidation of β-caryophyllene, and hence, not included in PRAM. The simulation results for yields from OH oxidation of β-caryophyllene, indicate that the MCM scheme is able to reproduce the experimental values. The mass yields derived from OH oxidation of isoprene vary from 0.01 - 0.31 covering a range of mass loadings from 0.003 - 132 $\mu g\ m^{-3}$. At low mass loadings < 10 $\mu g\ m^{-3}$ the maximum yield obtained is ~0.06, which is a factor of 2 greater than the experimental results obtained by Lee et al. (2006) and Kroll et al. (2005), where they measured yields of 0.02 and 0.01 – 0.03 respectively. Liu et al. (2016) measured higher mass yields (> 0.1) for similar loadings with a maximum upper limit yield of 0.15 for 22 $\mu g\ m^{-3}$, while our results under-predict the yields (~ 0. 10) for similar loadings. The OH oxidation of β-pinene and limonene results in a maximum yield of 0.28 for high mass loading of 319 $\mu g\ m^{-3}$ and 0.56 for loadings of 630 $\mu g\ m^{-3}$ respectively. These values are similar to the





measurements obtained by Lee et al. (2006) wherein they measured a yield of 0.31 for β-pinene SOA mass
loadings of 293 µg m$^{-3}$ and 0.58 for limonene SOA mass loadings of 394 µg m$^{-3}$. Yields for limonene SOA
mass loadings of 350 µg m$^{-3}$ are around 0.31 which is lower than the experimental values, measured by Lee et
al. (2006). The β-pinene SOA yields are comparatively well represented by MCM+PRAM in comparison to
the standalone MCM. On the other hand, the limonene mass yields are under-predicted by MCM+PRAM.

The OFR simulations results for the OH oxidation of BVOCs with an equivalent exposure range from

2.0 x 10$^{10}$ – 2.0 x 10$^{12}$ molecules cm$^{-3}$ s, is shown in Fig. 2. Friedman and Farmer (2018) found mass yields
of 7x10$^{-4}$ - 0.086 for α-pinene (ammonium sulfate seeded experiment), 0 - 0.12 for β-pinene (no seed
particles) and 0.0017 - 0.026 for limonene (no seed particles), by varying the OH exposures between 4.7 x
10$^{10}$ – 7.4 x 10$^{11}$ molecules cm$^{-3}$ s. Our simulated yields for OH oxidation of α-pinene (~0.05 - 0.31), β-
pinene (~0.019 - 0.2) and limonene suggest higher mass yields at both measured lower and higher values of
values of OH exposures and monoterpene concentrations (~ 12 – 270 ppb). In contrast to the experiments,
every simulation contained higher particle surface area, which could explain the higher simulated mass yields
for the compounds.

**Table 3.** Mass yields for OH oxidation of BVOCs at 293 K for different range of mass loadings using a chamber[†] and OFR[∥] setup.

| SOA mass loading (µg m$^{-3}$) | MCM + PRAM mass yields | MCM mass yields | BVOC | Experimental yields | References |
|---|---|---|---|---|---|
| 320[†] | 0.28 | 0.25 | β-pinene | 0.31 | Lee et al. (2006) |
| 350[†] | 0.31 | 0.06 – 0.11 | limonene | 0.58 | Lee et al. (2006) |
| 0-300[∥] | 0.05 – 0.31 | 0. – 0.2 | α-pinene | 7x10$^{-4}$ - 0.086 | Friedman and Farmer (2018) |
| 0-300[∥] | 0.019-0.2 | 0-0.13 | β-pinene | 0 - 0.12 | Friedman and Farmer (2018) |



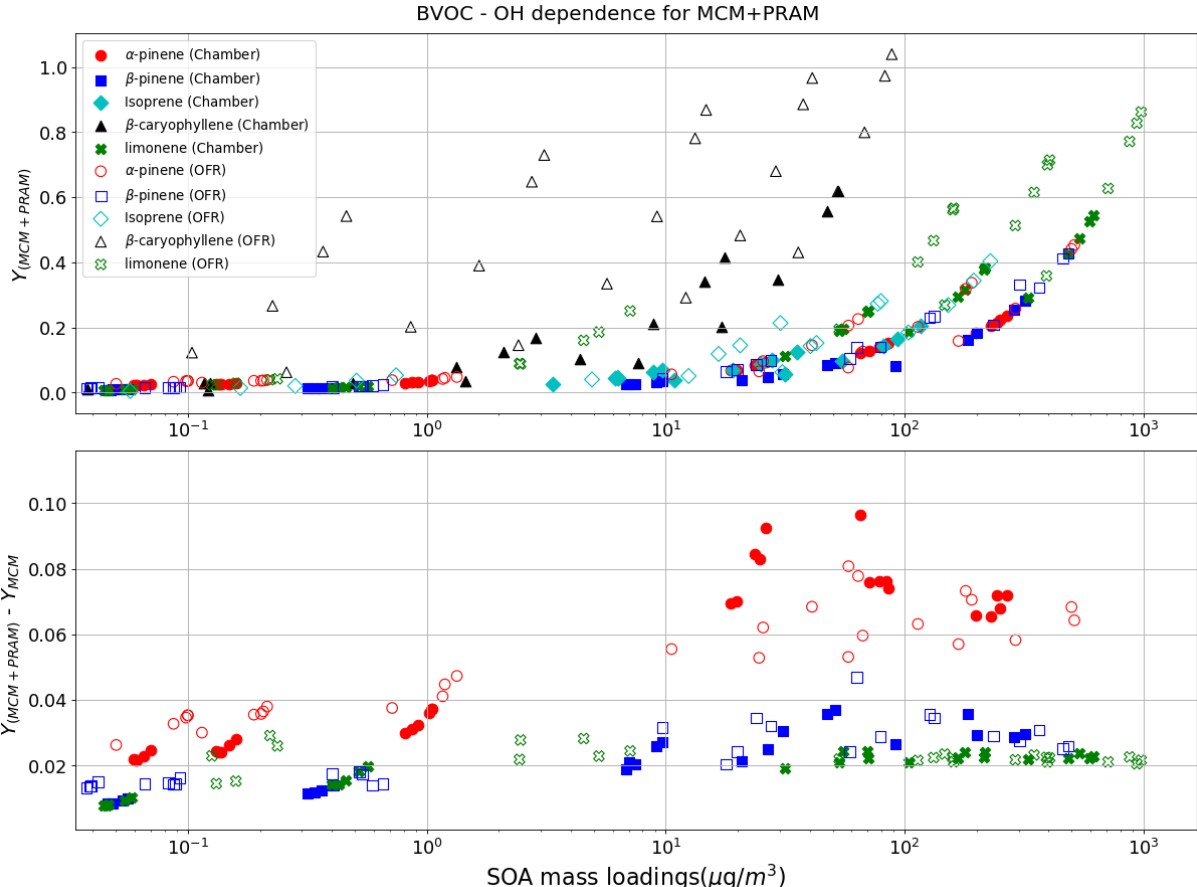

**Figure 2**. The mass yields from OH oxidation of BVOCs modeled after chamber and flow-tube settings. The figure shows a comparison of SOA mass yields obtained from application of MCM+PRAM and only MCM. The color scheme is the same as in figure 1. Additionally, isoprene has been included as it reacts with OH and produces considerable SOA yields.

**BVOC – NO₃ chamber and OFR simulations**

The yields obtained for oxidation of α-pinene (0.002-0.007) by NO₃ are low in comparison to those obtained by Nah et al. (2016), where they measured a yield of 0.036. Measured mass yields for limonene oxidation by NO₃ resulting in mass yields between 0.25-0.4 (Fry et al., 2011), whereas we obtain negligible (~0.0003) mass yields for the same. Due to limited experimental constraints, PRAM presently does not consider autoxidation of RO₂ formed from NO₃ oxidation of VOCs, which could explain the huge discrepancy between the measured and simulated mass yields.





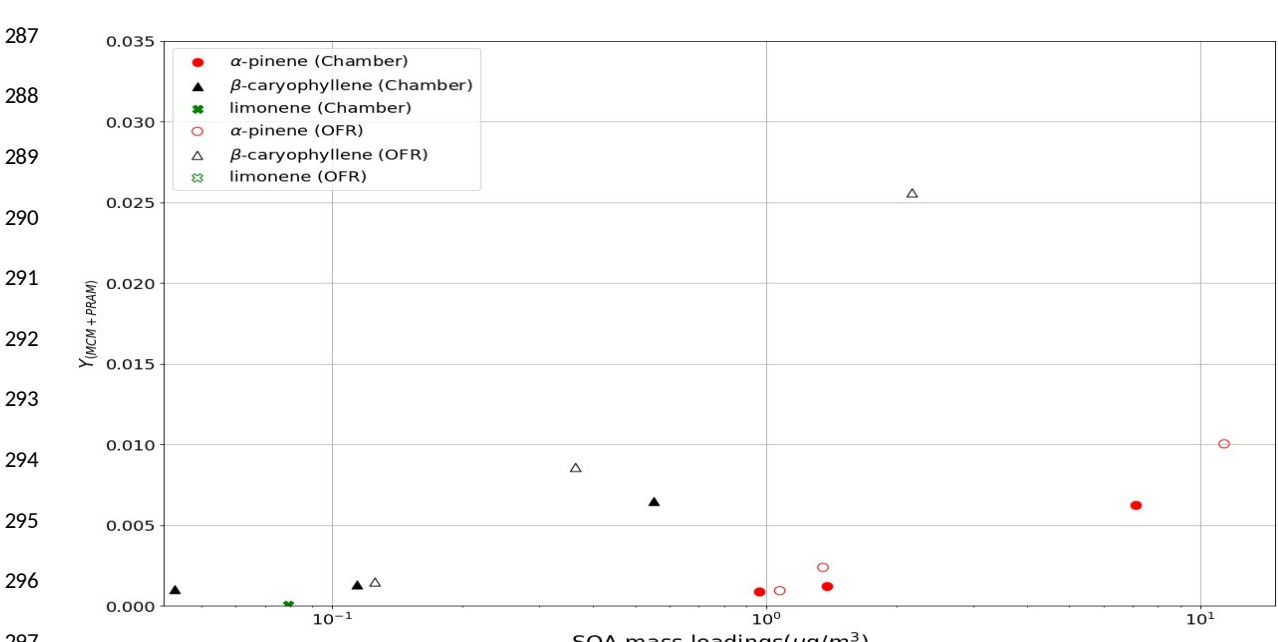

**Figure 3.** The mass yields from $NO_3$ oxidation of BVOCs modeled after chamber and flow-tube settings. The figure shows a
comparison of SOA mass yields obtained from application of MCM+PRAM. Appreciable mass yields were only obtained for α-
pinene, limonene and β-caryophyllene.
**3.3 $NO_x$ dependence**
Organic oxidation mechanisms are sensitive to $NO_x$ concentrations as they change the fate to $RO_2$
radical formed, thereby impacting the distribution of reaction products and aerosol formation (Presto et al.,
2005; Zhao et al., 2018). We modeled the SOA mass yields for α-pinene - $O_3$ setup with varying $NO_x$
concentrations (NO was varied whereas $NO_2$ was kept constant for all the runs), for initial α-pinene mixing
ratios in the range 0.5 - 200 ppb (Fig. 4). A maximum SOA yield value of 0.55 is obtained for a combination
of the lowest value of NO (0 ppb, red circles). As the NO concentrations increase from 0.2 ppb (blue squares)
to 5 ppb (green inverted triangles) the yields begin to decrease, and this pattern is observable and valid for all
concentration ranges of reacted precursor VOC. The $NO_x$ dependence of α-pinene ozonolysis is consistent
with the findings of Draper et al. (2015) and Presto et al. (2005) wherein they observed a trend of decreasing
SOA mass yields for α-pinene ozonolysis with increasing $NO_x$ concentrations. $NO_x$ concentrations alter the
$HO_2/RO_2$ ratio thereby impacting competing peroxy radical ($RO_2$) reaction pathways (Presto et al., 2005;
Sarrafzadeh et al., 2016).

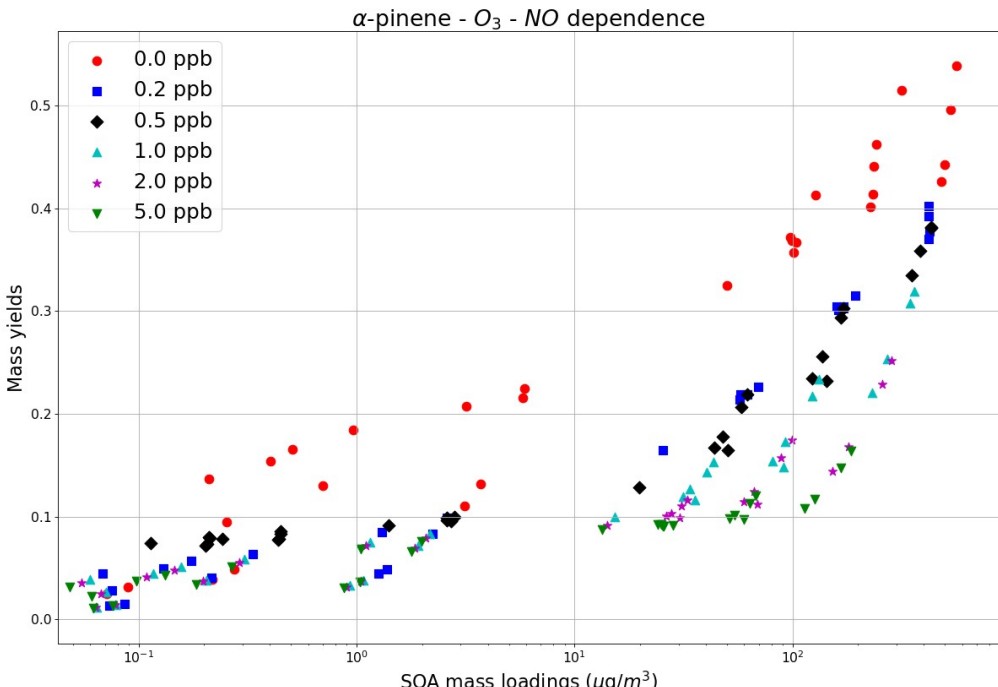

**Figure 4**. The SOA mass yields from O₃ oxidation of α-pinene modeled for different NO concentrations with the chamber setup. The model runs were performed using MCM+PRAM.

At low $NO_x$ concentrations $RO_2$ radicals undergo rapid autoxidation until they react with $HO_2$ or $RO_2$ resulting in production of low volatility hydro-peroxide products (Sarrafzadeh et al., 2016), closed shell monomers or dimers (Ehn et al., 201; Roldin et al., 2018), which increase SOA mass. This contrasts with high $NO_x$ conditions where the $RO_2$+NO reactions dominate over reactions with $HO_2$ or $RO_2$, resulting in the formation of more volatile products such as aldehydes, ketones and organonitrates (Presto et al., 2005; Sarrafzadeh et al., 2016), and likely suppressing the autoxidation process leading to a decrease in SOA mass loadings (Ehn et al., 2014).

Figure 5 shows the absolute contributions to SOA mass loadings by PRAM and MCM compounds at two different O₃ concentrations of 4 and 100 ppb and varying NO concentrations. The figure shows that with an increase in NO concentrations the contribution of PRAM compounds to the particle phase decreases at both 4 and 100 ppb of O₃ concentrations. In PRAM the $RO_2$ + NO reaction leads either to the formation of organonitrate HOM, closed shell monomers with carbonyl group or fragmentation products with higher volatility (Roldin et al., 2018). HOM Dimer formation is suppressed with increasing NO concentrations in





PRAM (Roldin et al., 2018) which explains the lower contribution by PRAM compounds to SOA mass
loadings with increasing NO. The PRAM contribution increases at low NO (<1 ppb) and then decreases
thereafter. At low NO concentrations (<1ppb), first generation $RO_2$ are capable of undergoing autoxidation
forming highly oxygenated $RO_2$ which subsequently reacts with NO forming organic nitrates (Ehn et al.,
2014). As NO concentrations exceed 1ppb the first generation $RO_2$ is scavenged by NO thereby reducing the
concentration of organonitrate HOM (Ehn et al., 2014), possibly affecting SOA yields. The MCM
contribution also decreases with increasing NO concentrations mostly due to the formation more volatile
organonitrates (Jenkin et al., 2019) .

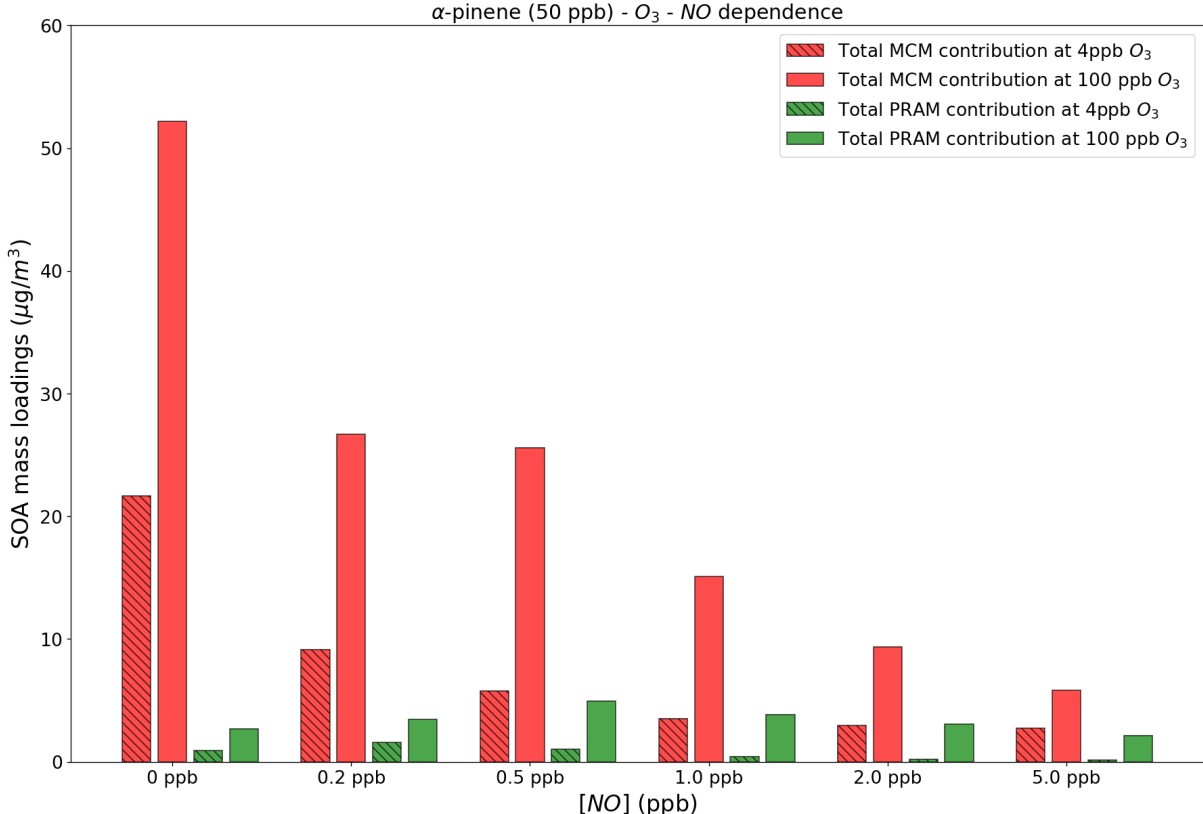

**Figure 5**. Contribution to the SOA mass loadings by total PRAM and MCM compounds at different $NO_x$ levels and $O_3$
concentrations. For comparison we use 4 ppb and 100 ppb $O_3$ concentrations, respectively, at 50 ppb α-pinene.
**3.4 Temperature dependence**

The formation of SOA from α-pinene ozonolysis in the temperature range of 258.15 - 313.15 K was

investigated in this study using MCM+PRAM. Strong dependence of SOA mass yield on temperature was
reported by Saathoff and Naumann, (2009) wherein they measured the decreasing mass yields from 0.42 at



273.15 K to 0.09 to 313.15 K for SOA loadings of 53 and 92  µgm⁻³ respectively. Our results in Figure 6
show increasing SOA mass yields for α-pinene ozonolysis with decreasing temperature, which is attributed to
the augmented condensation of oxidation products termed as semi volatile organic compounds (SVOC)
(Kristensen et al., 2017) at lower temperatures.

For α-pinene maximum mass loading < 150 µgm⁻³ the mass yields reach a maximum value of 0.38 at

temperatures as low as 258.15 K and decreases to 0.27 for a temperature of 293.15 K to 0.1 for the
temperature of 313.15 K. These yields are comparable to the results obtained by Kristensen et al.
(2017) where they measured yields of 0.39 for 258.15 K and 0.22 for 293.15 K for mass loading < 150 µgm⁻³.
The results show a weak dependence of SOA mass yields on temperatures in the range of 278.15 K - 313.15
K at low SOA mass loadings but become more pronounced as the mass loadings increase. At the lowest
temperature of 258.15 K the mass yields are higher in comparison to other temperatures regardless the mass
loadings. These results are in good agreement with the findings by Pathak et al. (2007) where they found a
strong temperature dependence of SOA mass yields at lower temperature (0 – 15º C), which decreases as the
temperature increases. Furthermore, similar to the measurements made by Pathak et al. (2007), our
simulations were able to reproduce the experimental findings that show no appreciable differences in the SOA
mass yields for loadings below 1 µgm⁻³ (initial mixing ratio of 1 ppb) for temperatures > 273.15 K.

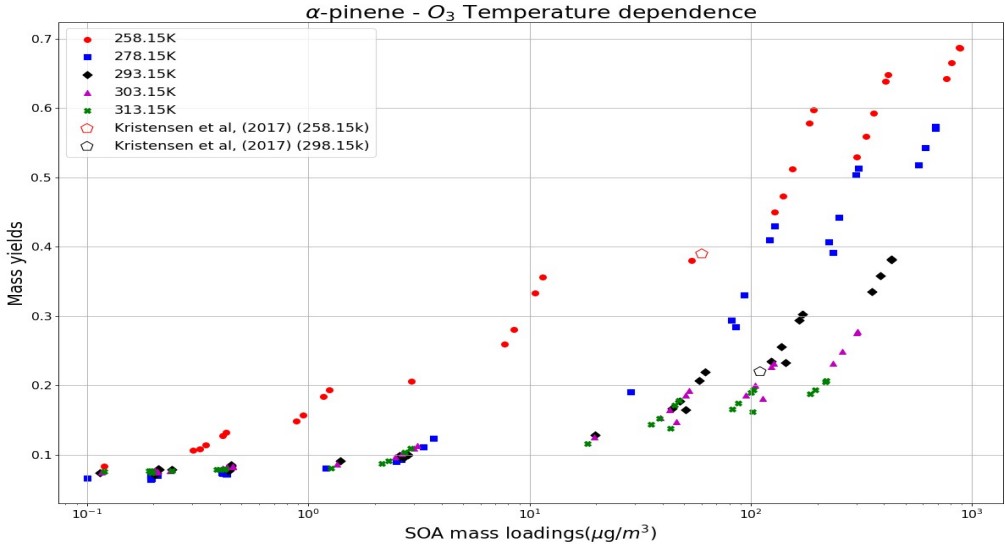

**Figure 6**. Temperature dependence of SOA mass yields at different temperatures using the MCM+PRAM. The open pentagons
represent measurement data from Kristensen et al. (2017) at 258.15 K and 298.15 K.





Figure 7 shows the volatility distribution of α-pinene ozonolysis derived SOA at different
temperatures. The saturation vapor pressure limits for defining extremely low volatility (ELVOCs - grey
shaded), low volatility (LVOCs - red shaded), semi volatile (SVOCs - green shaded) and intermediate
volatility (IVOCs - cyan shaded) organic compounds used in the Volatility basis set (VBS) are set according
to the values suggested in Donahue et al. (2012). In this work, we categorize compounds (ELVOCs, LVOCs,
SVOCs and IVOCs) based on effective saturation vapor pressures (C*) in the range of $\{10^{-5}$ to $10^3\}$ µgm$^{-3}$
and temperature of 298 K (Donahue et al., 2009). At the lowest temperature of 258.15 K, the SVOCs
contribution to the particle phase is dominant in comparison to LVOCs and ELVOCs, a trend which is
subsequently reversed as the temperatures are increased. At 293.15 K a majority of SVOCs and IVOCs are in
the gas phase while the contribution of LVOCs and ELVOCs to particle phases increases. These results are in
good agreement with observations made by Kristensen et al. (2017) wherein they observed an increasing
contribution of SVOCs at sub-zero temperatures of 258.15 K, which decrease the fraction of SOA formed
from ELVOCs. Again, it should be noted that the temperature dependence of peroxy radical autoxidation
product formation still needs further validation based on recent experiments (e.g. Quéléver et al., 2018).

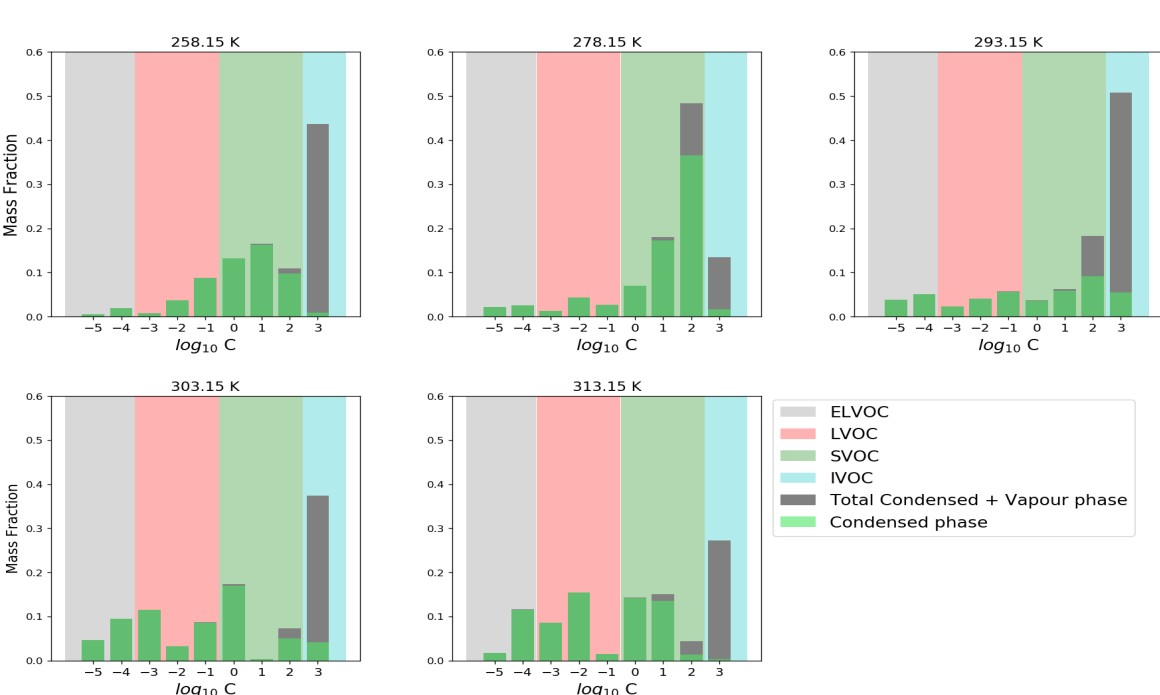

**Figure 7**. Modeled volatility distribution of SOA at different temperatures. The volatility bins span a range of effective saturation
vapor pressures C = C* = $\{10^{-5}$ to $10^3\}$ µgm$^{-3}$. The VBS distribution is based on a reference temperature of 298 K.





### 3.5 Composition

MCM+PRAM can be used to narrow down and compile a list of compounds playing a pivotal role in contributing to SOA mass loadings and, also compare the relative importance of implementing PRAM alongside the MCM. Figure 8 shows the most important compounds from both the MCM and PRAM that condense on the seed particles and have a dominant contribution (>95%) to the α-pinene ozonolysis SOA mass loading at 293 K.



**Figure 8**. MCM and PRAM Compounds contributing to > 95 % of SOA mass at 293 K and 50ppb $O_3$ and α-pinene concentrations.

At 293 K (Figure 8) the listed compounds (MCM+PRAM) contribute ~97 % to the SOA mass loading with PRAM compounds contributing to ~48 % (of 97 %) in comparison to MCM compounds which contribute ~52 % (of 97%). On lowering the temperature to 258K the relative contributions of PRAM drop to 15 % (of ~98





%), while MCM dominates by contributing ~85 % (of ~98 %) respectively (Figure S3a). The contribution of
PRAM increases to ~64 % (of ~97 %) and MCM contribution drops to 36 % (of ~97 %) at 313 K (Figure
S3b). These results reflect the importance of PRAM as its contribution plays an increasingly dominant role
with increasing temperatures and highlights the crucial few compounds that contribute to maximum SOA
mass loadings for α-pinene ozonolysis. The list of abundant compounds which together add up to contribute
more than 95 % of SOA mass loadings at 258 K, 293 K and 313 K are presented in the supplement Table 1s
(a, b & c). At 258 K MCM compounds namely pinonic acid ($C_{10}H_{16}O_3$, 4.4 %), C920PAN ($C_{10}H_{15}NO_7$, 9.3
%), C108NO3 ($C_{10}H_{15}NO_6$, 8.9 %), C811PAN ($C_9H_{13}NO_7$, 10.1 %), C717NO3 ($C_7H_9NO_6$, 11.3 %) contribute
significantly to the total SOA mass loadings while PRAM compounds such as $C_{10}H_{14}O_7$ (0.88 %), $C_{10}H_{16}O_4$
(1.3 %), $C_{10}H_{16}O_6$ (1.13 %) contribute significantly less. An increase in temperature to 293 K results in an
overall increase in contribution by PRAM compounds, with $C_{10}H_{14}O_{10}$ (3.6 %), $C_{10}H_{14}O_{11}$ (6.2 %), $C_{10}H_{16}O_{10}$
(3.2 %) playing an important role in contributing to the SOA mass loadings. This trend of relative increase in
the contribution by PRAM compounds over MCM compounds to SOA mass loadings is also evident as the
temperatures are further increased to 313 K, where the PRAM compounds $C_{10}H_{14}O_{11}$ (18.3 %), $C_{10}H_{14}O_{12}$ (6
%) and $C_{10}H_{16}O_{12}$ (6.6 %) play a dominant role in increasing SOA mass loadings.
**Conclusions**
We simulated SOA mass yields derived from the oxidation of various BVOCs (isoprene, α-pinene, β-
pinene, limonene and β-caryophyllene), by the oxidants $O_3$, OH and $NO_3$ using the zero-dimensional model
MALTE-Box. The gas phase chemistry was simulated using the MCM in conjunction with PRAM. The aim
was to verify the efficacy of MCM+PRAM in simulating the SOA mass yields. Additional simulations were
performed to test the MCM+PRAM under varying temperature and NO concentrations. A few important
compounds playing a major role in increasing the SOA mass yields for α-pinene ozonolysis at different
temperatures are also highlighted.
The simulations were designed to resemble ideal smog chambers experiments and experiments in
oxidative flow reactors (OFR). No interactions between the gas phase and chamber walls were considered
during the simulations. For the smog chamber setting, the standalone MCM generally under-predicts the mass
yields obtained by the ozonolysis and OH oxidation of BVOCs. In contrast, the yields derived using
MCM+PRAM for the smog chamber setup is in good agreement with the experimental results. For an
idealized OFR setup, MCM+PRAM over-predicts the yields in case of ozonolysis and OH oxidation of
BVOCs, while again the MCM under-predicts the SOA yields. This over prediction in mass yields for the
OFR simulation was a result of higher CS used in the OFR. The relative contribution of HOM monomers and



dimers to the particle phase in OFR simulations is low when compared to the chamber simulations. This is
due to higher $RO_2$ concentrations in OFR leading to termination of peroxy radical autoxidation, thereby
affecting SOA yields. This needs to be considered when applying yields based on OFR simulations in
regional or global chemical transport models

The BVOCs included in this study do not produce appreciable SOA mass yields when oxidized with

$NO_3$, as PRAM currently does not consider autoxidation of $RO_2$ formed from $NO_3$ oxidation of VOCs. This
underlines the need for developing a $NO_3$ oxidation scheme which can better constrain and predict SOA mass
yields. In accordance to the previous studies, the simulated SOA yields tend to decrease at higher
temperatures. The PRAM contribution to mass yields at low temperatures (258.15 K) is ~14 %, which is
substantially lower in comparison to MCM (~86 %). As the temperature is increased to 313.15 K, the
contribution of PRAM to SOA mass yields begins to dominate over MCM. This most likely is due to MCM
containing more SVOCs (compounds classified as SVOCs at 298 K), which show stronger contribution to
particle phase at lower temperatures, due to decrease in saturation vapor pressures with temperature.  It should
be noted that the present temperature dependency of mass yields using PRAM are a first, and currently the
best estimate in understanding the influence of temperature on the peroxy radical autoxidation formation. The
simulated SOA yields with varying NO concentrations agree well with experimental results, i.e. SOA yields
decrease with increasing NO concentrations due to the formation of more volatile compounds such as
organonitrates and ketones.

Using PRAM in addition to MCM has paved way to bridge the gap in understanding the role and

contribution of peroxy radical autoxidation to SOA formation. The variation of SOA yields for temperature
and NO concentrations, indicates the limitations of global and regional models in predicting e.g. cloud
condensation nuclei (CCN) effects using fixed SOA yields. The good agreement of modeled and experimental
yields from smog chambers, respectively, could further help us parameterize the SOA yields, that could be
applied at a global and regional model scale, to more accurately predict the direct and indirect impact of
aerosol particles on e.g. radiation balance by aerosol scattering/absorption and CCN concentrations.

### Author Contributions

CX and MB served as the chief authors and editors of the paper. CX was performing the model simulations.
The study was designed by CX, MB and PR. All other co-authors contributed to the analysis and writing of
the paper.





**Acknowledgements**

The presented research has been funded by the Academy of Finland (Center of Excellence in Atmospheric
Sciences) grant no. 4100104 and the Swedish Research Council FORMAS, project no. 2018-01745.

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
