# Peer review of "Aerosol Mass yields of selected Biogenic Volatile Organic Compounds – a"

_Atmospheric Chemistry and Physics, 2019_

## Editor Comment (EC1) · Barbara Ervens (Editor) · 28 Jun 2019

The work relies heavily on the detailed mechanism PRAM (peroxy radical autoxidation mechanism, Roldin et al., 2018) you refer to several times. Unfortunately, 'Roldin et al., 2018' is not listed in the reference list; instead (?) 'Roldin et al., 2019 in review' is listed but not cited.

In order to make it easier for reviewers and other readers to evaluate your study in light of the underlying mechanism, it is essential to have access to the cited paper or, at least, to the mechanism as a separate source (e.g. a link to a website).

[Figure]

Please (i) clarify the mix-up in the references and (ii) give detailed information on PRAM in a response to this comment.

---

## Author Comment (AC1) · 12 Jul 2019

Dear Editor Barbara,

We apologize for mixing up the Roldin 2019 reference with a wrong date (2018), but have corrected this in the new version of the manuscript. We also have added a detailed description of the PRAM mechanism and two references (Öström et al., 2017 and Ximeng et al., 2018) which applied the same mechanism but without mentioning PRAM as the name came later.

The reason for the mixing of the year in the reference Roldin at al. is based on the

fact that we have submitted the Roldin paper already in the end of last year to Nature communication but for several reasons (e.g. Christmas holidays, problem to find reviewers, ..) we only got the outcome of the review a couple of weeks ago. Recently Pontus Roldin has send the feedbacks to the reviewers and of course we hope to get this paper published asap. So, when we submitted this paper we believed that Roldin et al. 2019 would be already in press or at least accepted but now all seems to be delayed. The Roldin et al. paper includes the full chemical mechanism of PRAM and to avoid a double publication of PRAM in two different journals we have not added the chemical code in Xavier et al. 2019.

Best Regards, Carlton Xavier and co-authors.

---

## Referee Comment (RC1) · Anonymous Referee #1 · 23 Jul 2019

This paper presents a study of the SOA-forming potential of autoxidation mechanisms for several important BVOCs, as expressed in the PRAM chemical mechanism. A suite of box model simulations is performed for both chamber and OFR conditions using the MCM alone and in combination with PRAM. SOA yields simulated using MCM+PRAM show significantly better agreement with experimental data than do the MCM-only simulations, indicating the importance of the autoxidation reactions included in PRAM to SOA production. Sensitivity studies are also presented showing of the influence of temperature and NO variations on the contribution of the autoxidation mechanism to the overall SOA yield. This appears to be a careful and comprehensive study, and is a valuable contribution to the literature. I recommend publication after the following

points have been addressed.

General comments:

1. As noted by the Editor, it would be extremely helpful to be able to view the Roldin manuscript describing PRAM. I must leave it to the editor's discretion whether publication of the present manuscript should be contingent on publication of Roldin et al (2019). References to Roldin (2018) should be corrected to Roldin (2019) throughout.

2. The discussion is pertinent and interesting but is also convoluted in places and difficult to read. It would benefit from a careful re-writing for language clarity and brevity. For example, lines 243-245 read: "Similarly, the current lack of peroxy radical autoxidation product mechanism for b-caryophyllene and isoprene result in Delta-Y=0 values for PRAM." How about saying something simpler, like: "Peroxy radical autoxidation reactions of b-caryophyllene and isoprene OH products are currently not included in PRAM, so the mechanisms are not compared in these cases (Fig 2b)."?

3. Comparisons with many published experimental results are cited in the text but are not included in the Tables (especially results where no PRAM is yet available). This reviewer suggests that it would be extremely helpful to move these simulation/data comparisons to the yield tables and figures (whether as points or as ranges). Then the agreement (or otherwise) could be summarized in the text without having to list all the specific numbers. This would make the text and its arguments easier to follow.

4. Perhaps I am missing something, but if the peroxy autoxidation reactions are not available for certain species/oxidant combinations, wouldn't it be more correct and less confusing to call the mechanism for those species/oxidant combinations MCM (or MCM-only) instead of MCM+PRAM or PRAM? (Throughout the manuscript, including Figure captions).

5. Please explain whether some of the SOA formed in MCM is converted to different species in PRAM? Put another way, is the SOA formed in PRAM completely additional

to that formed in MCM, or is there some conversion as a result of the autoxidation? If the latter, please discuss the level of "double-counting" of products in the MCM/PRAM side-by-side mass spectra figures.

6. The range of sensitivity conditions seems rather wide: the temperature extremes are beyond usual ambient temperatures. Please discuss whether the results from the extreme cases are likely to be environmentally or observationally relevant.

7. Given that SOA yields in an OFR are sensitive to particle surface area, several points arise. i) The Abstract states that MCM+PRAM overestimates OFR yields and gives increased particle surface area as the reason. The casual ("abstract-plus-figures") reader is left wondering why the simulations didn't use the same particle loadings as the literature. Is it possible to provide a little context in the abstract, to explain? ii) Does the over-prediction of modeled OFR SOA suggest that the literature experiments in the comparisons used too little seed to obtain stable yields? iii) The MCM+PRAM OFR overestimation is not readily apparent from Tables 2 and 3. It would be helpful to include lines that compare the results with measurements under the same loadings if possible, so the disagreement is more apparent to the reader.

8. The earlier termination of the autoxidation mechanism in the OFR cases is attributed to "The higher absolute RO2 concentrations in the OFR simulations …. ie the high RO2 concentrations in the OFR cause termination of the peroxy radical autoxidation chain before the RO2 become highly oxygenated…." (line 229 ff.) This disagrees with the conclusions of Peng et al 2019 (https://doi.org/10.5194/acp-19-813-2019, 2019) who found that "for most types of RO2, their bimolecular fates in OFRs are mainly RO2+HO2 and RO2+NO, similar to chambers and atmospheric studies." At low NO, the high concentration of HO2 in the OFR leads to more rapid RO2 loss; at high NO, RO2+NO makes RO2 lifetime very short in the OFR. Please discuss whether the current modeling analysis is consistent with that work.

Specific line-by-line comments:

Line 29: The scale of SOA contribution... is "still" subjected to high uncertainties. Is there a more recent reference than 2011?

Line 41: Does this mean Ehn (2014)? Ehn (2012) is not listed in the references.

Line 57: Are there any measured O/C ratios in relevant systems that could be compared with?

Line 94: The timing seems confused: MCM+PRAM (Damian et al 2002) vs PRAM (Roldin et al 2018). Please clarify. (Did Damian et al really refer to MCM+PRAM?)

Line 98: What sort of fraction of the peroxy radicals is considered in PRAM?

Line 115-118: Please explain whether there is likely to be any bias from using two different systems to estimate p0 for different species subsets?

Line 188: I think the phrase "contribution to SOA mass" is misleading. It suggests proportion of the SOA made up by species "i", whereas the figure actually shows "SOA mass yield".

Lines 217 & 221, and in general: When referring to "our model" It would be helpful to distinguish at that point which version is being used in each case (MCM+PRAM or MCM), so that the reader is reminded whether or not PRAM is being used. The distinction is made a few sentences later: a little reorganization would help this discussion.

Line 220: The values quoted in the text for OFR-simulation SOA yield from a-pinene ozonolysis do not match the values quoted in Table 2. Why the discrepancy?

Line 240: This section needs an easier-to read introductory sentence.

Line 243 and following: Please list in Table 3 the experimental results of Kristensen 2017 and others cited in Section 3.2.

Line 244: It's not really true that Y=0 in these cases, since Y is the result of a comparison, and here there is nothing to compare (since there really isn't a "PRAM" for these

species-oxidant combinations).

Line 250: It's usual to say that the model results are in good agreement with previous measurements, not the other way around.

Line 274: Please briefly remind the reader why the simulations used more surface area than the experiments? It seems to be an important factor in the disagreement.

Line 297: Section number is missing (3.3). Subsequent section numbers should be updated accordingly.

Line 302: This sentence is difficult to make sense of. Is this what is meant? "Varying NOx concentrations changes the fates of RO2 radicals formed during organic oxidation, thereby impacting . . ."

Line 330: Please be specific that the PRAM contribution increases with increasing NO for NO < 1ppb. (It's not quite clear the way it's currently written.)

Line 382: I suspect this means that the compounds shown in Fig 8 contribute >95% to the SOA mass loading when summed in decreasing order of contribution. Please clarify the text. (It's said better in the caption and in Line 385.)

Line 423: I think this means to say something like "We do not simulate appreciable mass yields from the oxidation of BVOCS with NO3". The current text claims to describe the behavior of the actual compounds, but I think it really intends to describe their behavior in the model. It's an important distinction.

Comments on the Figures:

1. Why are the points in the Figures arranged in clumps/streaks? Please explain early on.

2. The caption to Fig 1 says ". . . from simulations with MCM+PRAM and PRAM." Shouldn't it really say "MCM+PRAM and MCM"?

[Figure]

3. Figs 1 & 2: Please denote the two panels 'a' and b' and refer to them that way in the caption and text. This would help the reader and might help clarify the flow of the discussion.

4. Figs 1 & 2: Please add notes to the figure captions to clarify which species are omitted from the comparison in each case (i.e. which species use MCM-only)

5. Figs 1 & 2, 2nd panels: It is sometimes hard to figure out whether MCM makes any contribution at all. Please make this clearer by either a) plotting the ratio YMCM / Y(MCM+PRAM) instead of (or in addition to) the difference between the two, or at least b) using the same gridline interval in both panels.

6. Please mention Figure 3 somewhere in the text, or remove it.

Comments on the Tables:

1. Table 2: what are the figures in parentheses in the Experimental Yields column? Why are they not always present? Please explain.

2. Table 2: Please include the b-pinene and b-caryophyllene MCM-to-literature comparisons mentioned in the text (lines 203-212).

3. Table S1a: To make this information easier to digest, I suggest listing the compounds in descending order of contribution to SOA mass. Also: Is this just the a-pinene ozonolysis case? Please clarify. If it's for various precursors, please indicate which precursor is relevant for each product.

Minor Language Editing Suggestions:

Line 29: "… is still subject to …" (Not "subjected")

Line 134: "by contrast" might be a better phrase than "on the contrary"

Line 183 (suggestion): Move header for Section 3.1 to after "flow-tube experiments" (in line 187)

Line 240: Replace "resulting" with "result".

Line 243: the word "Similarly" seems strange here. (It would usually be understood to refer to the previous sentence). Perhaps this means: "As in the ozonolysis case"?

Line 283: "Due to limited experimental constraints, PRAM presently does not consider autoxidation of RO2 formed from NO3 oxidation of VOCS". I suggest that moving this information to the top of the paragraph would help the reader more quickly make sense of the comparisons presented.

Line 311 almost duplicates Line 302. It would be good to combine these two sentences, for brevity.

Line 335: ". . . the formation OF more volatile . . ." (add word "of")

Line 348: ". . . and decrease (no 's') to 0.27 at 293.15K AND to 0.1 at 313.15K" (add word "and")

Line 351-2: Maybe this is "a weak dependence . . . . WHICH becomes more pronounced. . ."? ("which", not "but")

Line 385 duplicates some of lines 381-383. Please condense.

Line 428: (suggestion) "substantially lower than that of MCM"

Line 430: MCM *produces* more SVOCs, it doesn't just "contain" them.

Line 441: delete word "respectively"

Line 473" "has paved THE way" (add word "the"). Or substitute something simpler like "helps us".

―――――――――――――

---

## Referee Comment (RC2) · Anonymous Referee #2 · 6 Aug 2019

This is a good informative study that compares MCM and MCM+PRAM mechanisms to derive model capabilities of known peroxy radical autooxidation mechanisms.

Following are some comments that are recommended to improve the current work: 1. Add a section on experimental details before model description. This could be just a summary of various experimental studies the work is using to evaluate the model with justifications for why they were chosen.

2. On page 17 the authors mention T-dependence of peroxy radical autotoxidation needs further improvement/validation. More details on their assumed T-dependence in PRAM are needed. For example, what was the assumed T-dependence as a function

of precursor VOC, oxidant, NO etc.? What was the T-dependence of saturation vapor pressures of SVOCs in Figure 7? Seems there are 2 different T-dependence that need to be explicitly stated: (A) T-dependence of autooxidation chemistry (B) T-dependence of their C* i.e. saturation vapor pressure or a physical process of gas-particle partitioning

This is T-dependence is a very important part and needs to be discussed clearly. Also discuss measurements of such T-dependencies as applicable.

(3) Figure 7: Would it be possible to start with a VBS fit at 313 K, and then derive the VBS fit at 258K or vice versa with these T-dependencies without having to run MCM+PRAM at each of these temperatures? This is important for regional and global models that rely on VBS and cannot run full MCM+PRAM.

(4) Do the authors have any recommendations for condensed versions of MCM+PRAM that could be used in regional and global models to predict SOA yields and their oxidation state?

---

## Editor Comment (EC2) · Barbara Ervens (Editor) · 9 Aug 2019

I have received a late reviewer comment; see below. I would like to ask you to address it, too, during the revision of your manuscript as it will help improving and clarifying your study.

Reviewer comment: My remaining criticism, which would still lead to rejection for me (or contengency on the publication of their Nature Comms paper), is that the mechanism still not listed in their revision. Moreover, of the 2 references they have stated are in the bibliography that describe PRAM in detail, the Öström one is incomplete and the Ximeng one is missing. I did find a PRAM model listing in Emilie Öström's thesis online

(https://portal.research.lu.se/portal/files/31835271/Emilie_str_m_hela_inkl._omslag.pdf), but it does not seem to correspond to what is in the current Xavier et al. paper, and paper III in the thesis may refer to an old version of what later became the Nature Comms submission. I do not feel that the model description in sufficient to allow publication with the current existing confusion.

---

## Author Response (AR1)

**Comments for the Reviewers**

We thank the reviewers for the positive and constructive comments. The author responses are in blue.

**Late reviewer comments.**

Reviewer comment: My remaining criticism, which would still lead to rejection for me (or contengency on the publication of their Nature Comms paper), is that the mechanism still not listed in their revision. Moreover, of the 2 references they have stated are in the bibliography that describe PRAM in detail, the Öström one is incomplete and the Ximeng one is missing. I did find a PRAM model listing in Emilie Öström's thesis online (https://portal.research.lu.se/portal/files/31835271/Emilie_str_m_hela_inkl._omslag.pdf), but it does not seem to correspond to what is in the current Xavier et al. paper, and paper III in the thesis may refer to an old version of what later became the Nature Comms submission. I do not feel that the model description in sufficient to allow publication with the current existing confusion.

Thank you for being patient. The Nature Communications paper by Roldin et al., 2019 has now been accepted for publication and will be published on 25$^{th}$ September 2019. In this paper there is a link provided to download the complete PRAM mechanism written in a format compatible with the Kinetic PreProcessor (KPP) together with all species information [https://doi.org/10.1594/PANGAEA.905102]. Also the reference to Roldin et al., 2019 will be updated in case this paper has been successfully accepted.

This information is also provided in this paper under 'Data availability' in the end of the paper as following text:

The complete PRAM mechanism written in a format compatible with the Kinetic PreProcessor (KPP) together with all species information can also be downloaded from https://doi.org/10.1594/PANGAEA.905102

**Reviewer 1**

This paper presents a study of the SOA-forming potential of autoxidation mechanisms for several important BVOCs, as expressed in the PRAM chemical mechanism. A suite of box model simulations is performed for both chamber and OFR conditions using the MCM alone and in combination with PRAM. SOA yields simulated using MCM+PRAM show significantly better agreement with experimental data than do the MCM-only simulations, indicating the importance of the autoxidation reactions included in PRAM to

SOA production. Sensitivity studies are also presented showing of the influence of temperature and NO

variations on the contribution of the autoxidation mechanism to the overall SOA yield. This appears to be a careful and comprehensive study, and is a valuable contribution to the literature. I recommend publication after the following points have been addressed.

Thank you

RC1. As noted by the Editor, it would be extremely helpful to be able to view the Roldin manuscript describing PRAM. I must leave it to the editor's discretion whether publication of the present manuscript should be contingent on publication of Roldin et al (2019). References to Roldin (2018) should be corrected to Roldin (2019) throughout.

Reply. Thank you for the comment. We have now changed the reference Roldin (2018) →  Roldin (2019)

throughout the manuscript. We also have added a detailed description of the PRAM mechanism (L 98-112)

and two references (Öström et al., 2017; Qi, 2018)  which applied an earlier version of PRAM, though not using the acronym PRAM as the name came later.  And as mentioned above the manuscript by Roldin et al.,

2019 has been accepted for publication and PRAM from Roldin et al., 2019 can also be found at the following link https://doi.org/10.1594/PANGAEA.905102.

RC2. The discussion is pertinent and interesting but is also convoluted in places and difficult to read. It would benefit from a careful re-writing for language clarity and brevity. For example, lines 243-245 read:

"Similarly, the current lack of peroxy radical autoxidation product mechanism for b-caryophyllene and isoprene result in Delta-Y=0 values for PRAM." How about saying something simpler, like: "Peroxy radical autoxidation reactions of b-caryophyllene and isoprene OH products are currently not included in PRAM, so the mechanisms are not compared in these cases (Fig 2b)."?

Reply. Thank you for bringing this to notice. We agree that parts of the manuscript will benefit from the use of simpler language for clarity. The line has now been omitted. We have rephrased the paragraph and the above line has been changed. Line omitted →

Line added (266-268) →  Currently there are no experiments providing HOM yields from OH oxidation of β- caryophyllene, and hence, those species are not included in PRAM.

Line added (269-270) → Only MCM was used for modeling the mass yields for OH oxidation of isoprene due to current lack of PRAM mechanism for isoprene.

RC3. Comparisons with many published experimental results are cited in the text but are not included in the Tables (especially results where no PRAM is yet available). This reviewer suggests that it would be extremely helpful to move these simulation/data comparisons to the yield tables and figures (whether as points or as ranges). Then the agreement (or otherwise) could be summarized in the text without having to list all the specific numbers. This would make the text and its arguments easier to follow.

Reply. Thank you for the comment. I agree that moving the comparison between model to experiment values to the tables and figures can make it easier to follow arguments. Table 2 (a and b). and Table 3. have been updated to include comparison values between MCM+PRAM, MCM and experimental values for all compounds. Additionally Figures 1 and 2 have been replaced with four figures (Fig. 1, 2, 3 and 4). PRAM is currently not available for the ozonolyis of β-caryophyllene and β-pinene and for OH oxidation of β-caryophyllene and isoprene. Hence, we separated the figures to show how applying MCM+PRAM improves the mass yields estimation in comparison to only MCM compounds. Figure 1 and 3 show the improvement to mass yields by ozonolyis and OH, respectively for species where PRAM is available, whereas Figure 2. and 4. show the performance of MCM for oxidation of species not currently included in PRAM. Experimental values have also been added to the Figures 1,2 & 3 for better comparison. Changes to the text have been made at lines

L207-210→ The mass yields derived using MCM+PRAM for α-pinene ozonolysis are in good agreement with the experimental yields measured for similar mass loadings by Kristensen et al. (2017) and Pathak et al. (2007). The standalone MCM, on the other hand, severely under-predicts the mass yields for α-pinene ozonolysis.

L265-269→For β-caryophyllene, the modeled values are in good agreement with experimental measured yields in the range of mass loadings provided by Griffin (1999) and Tasoglou and Pandis (2015). Currently there are no experiments providing HOM yields from OH oxidation of β-caryophyllene, and hence, not included in PRAM

RC4. Perhaps I am missing something, but if the peroxy autoxidation reactions are not available for certain species/oxidant combinations, wouldn't it be more correct and less confusing to call the mechanism for those species/oxidant combinations MCM (or MCM-only) instead of MCM+PRAM or PRAM? (Throughout the manuscript, including Figure captions).

Reply. Thank you for pointing this out. You are right that it would be less confusing to refer to species/oxidant combination where no PRAM is available as MCM only. We have now corrected this throughout the manuscript.

RC5. Please explain whether some of the SOA formed in MCM is converted to different species in PRAM? Put another way, is the SOA formed in PRAM completely additional to that formed in MCM, or is there some conversion as a result of the autoxidation? If the latter, please discuss the level of "double-counting" of products in the MCM/PRAM side-by-side mass spectra figures.

Reply. Thank you for this comment. I assume you mean to ask if the compounds formed in PRAM are additional to compounds formed in MCM. Yes, the compounds formed in PRAM are additional to the compounds formed in MCM, ensuring that in the complete MCM+PRAM mechanism the total number of carbon atoms is conserved starting from the initial precursor. E.g. if you sum up the first generation $RO_2$ formed from α-pinene ozonolysis 91 % will react further in the MCM mechanism and the remaining 9 % will undergo autoxidation in PRAM. If only MCM chemistry is used in the model 100 % of the formed first generation $RO_2$ will continue in the MCM chemistry. Hence, when we implement PRAM together with MCM, the amount of MCM oxidation products are decreased. The most extreme case is limonene where 21.9 % of the first generation $RO_2$ go to PRAM and only 78.1 % follow the MCM chemistry (see Supplementary Table 1 in Roldin et al., 2019). An extract from Roldin et al., 2019:

*"When α-pinene is oxidized by $O_3$ one of the two ring structures is broken but a cyclobutyl ring is left intact in the $RO_2$ isomers ($C10H15O4$ ) that are formed. According to quantum chemical calculations by Kurtén et al.,[20] the cyclobutyl ring inhibits multiple autoxidation steps and prevents the first intramolecular H-shifts reactions rates to exceed 0.3 $s^{-1}$ at 298 K. In PRAM we therefore assigned a rate constants of 0.3 $s^{-1}$ at 298 K for the first H-shift reaction (R11 in SI Table S4). Kurtén et al. also examined possible reaction pathways that can lead to opening of the cyclobutyl ring. According to Kurtén et al. the ring opening can likely occur via alkoxy radicals. Such pathways are also present in MCMv3.3.1 when the $RO_2$ isomer, with the MCM name C107O2, react with NO, $NO_3$ or other $RO_2$ and form an alkoxy radical C107O that can isomerize and react*

*with $O_2$ and form a $C10H15O5$ peroxy radical named C108O2. In PRAM we therefore included the possibility of such additional HOM formation pathway for α-pinene, which is initiated by the reaction between C107O2 and other $RO_2$ (R1152 followed by R20 and R12-R19 in SI Table S4)."*

RC6. The range of sensitivity conditions seems rather wide: the temperature extremes are beyond usual ambient temperatures. Please discuss whether the results from the extreme cases are likely to be environmentally or observationally relevant.

Reply. Thank you for this interesting comment. We have used 2 temperature extremes in the simulation of SOA mass yields, 258 K and 313 K respectively. Measurements have shown high concentrations of SOA in the free troposphere around 2-6.5 km (Heald et al., 2005). The lower temperature extreme of 258 K is a good approximation for the free troposphere. Also, there have been a multitude of chamber experiments performed at these extreme temperatures (both at 258 and 313 K) (Kristensen et al., 2017; Saathoff and Naumann, 2009) and the aim of this study was to check the efficacy of MCM+PRAM in estimating yields at varying temperatures.

RC7. Given that SOA yields in an OFR are sensitive to particle surface area, several points arise. i) The Abstract states that MCM+PRAM overestimates OFR yields and gives increased particle surface area as the reason. The casual ("abstract-plus-figures") reader is left wondering why the simulations didn't use the same particle loadings as the literature. Is it possible to provide a little context in the abstract, to explain? ii)Does the over-prediction of modeled OFR SOA suggest that the literature experiments in the comparisons used too little seed to obtain stable yields? iii) The MCM+PRAM OFR overestimation is not readily apparent from Tables 2 and 3. It would be helpful to include lines that compare the results with measurements under the same loadings if possible, so the disagreement is more apparent to the reader.

Reply. Good question.

i) We have tested different particle surface area scenarios and subsequently chosen an area at which the yields estimation are not surface area limited (Supplement figure S1). On using low particle surface area we did not produce comparable particle mass loadings. Using the current particle surface area (corresponding to CS value of 0.067 s$^{-1}$) we are able to simulate mass loadings comparable with literature. We have now modified the comparison section where we compare the mass yields with similar particle loadings from literature. After re-examining the comparison of yields between experimental and simulations for similar particle loadings we find that for OH oxidation of α-pinene (Bruns et al., 2015) and β-pinene, mass yields are in good agreement with experimental values, whereas mass yields from OH oxidation of limonene are higher at similar particle loadings (Table 3). For ozonolysis, β-pinene mass yields are drastically under-predicted, while we see that α-pinene mass yields are in good agreement with results from Kang and Root (2007), after taking into account that the mass yields increase by a factor of 1.4 on adding seed particles. Changes to the abstract have now been made as follows:

→ Compared to experimental yields, the OFR simulations using MCM+PRAM yields were in good agreement for BVOCs oxidized by both O$_3$ and OH. On the other hand, a standalone MCM under-predicted the SOA mass yields.

ii) Yes.  Ahlberg et al., (2019), Lambe et al., (2015) and Kang and Root (2007) have all found an increase in SOA mass yields when seed particles were used. The experiments by Friedman and Farmer (2018) have also measured lower particle surface area leading to an underestimation of SOA yields. The extract from

Friedman and Farmer (2018)

*"The SMPS utilized in our study detected a maximum particle diameter of 289 nm; this upper limit may lead*

*to an underestimation of the total particle mass for particles growing to sizes larger than 289 nm and lower*

*reported yields compared to other studies utilizing a larger particle size range.*"

iii) The values to compare the simulation results with experiments have now been added to tables 2 and 3.

Again we re-iterate that on re-examining the simulation and experimental values for similar particle loading we find that the mass yields generally agree well the experimental values (point (i)).

RC8. The earlier termination of the autoxidation mechanism in the OFR cases is attributed to "The higher absolute RO2 concentrations in the OFR simulations . . .. I.e. the high RO2 concentrations in the OFR cause termination of the peroxy radical autoxidation chain before the RO2 become highly oxygenated. . .." (line 229

ff.) This disagrees with the conclusions of Peng et al 2019 (https://doi.org/10.5194/acp-19-813-2019, 2019)

who found that "for most types of RO2, their bimolecular fates in OFRs are mainly RO2+HO2 and RO2+NO, similar to chambers and atmospheric studies." At low NO, the high concentration of HO2 in the OFR leads to more rapid RO2 loss; at high NO, RO2+NO makes RO2 lifetime very short in the OFR. Please discuss whether the current modeling analysis is consistent with that work.

Reply. The study conducted by Peng et al., (2019) focuses on OH dominated atmospheres. The high concentrations of $RO_2$ described in the above sentence focuses on $O_3$ dominated atmosphere. Figure C1 shows that α-pinene – OH oxidation forms fewer dimers compared to ozonolysis. Figure C2 shows that our results are actually in agreement with the results from Peng et al., (2019). Compounds such as C10H18O5 (1.6),

C10H18O6 (1.3), C10H18O7 (1.1), C10H18O8 (0.5) and C10H18O9 (0.2) are products of $RO_2 + HO_2$

reaction in PRAM (Roldin et al., 2019). The contribution of the above mentioned compounds are higher compared to the dimer contribution to SOA mass loadings in a OH dominated atmosphere, thereby supporting the argument that $RO_2 + HO_2$ reaction pathway dominates over $RO_2 + RO_2$ pathways in OH initiated and dominated atmospheres.

[Figure]

Figure C1. The upper panel shows the mass spectra for OFR simulations performed for α-pinene OH oxidation, while the lower panel shows the mass spectra for α-pinene ozonolysis.

[Figure]

Figure C2. Compounds contributing to SOA mass loadings for α-pinene OH oxidation using an OFR setup.

Line 30: The scale of SOA contribution. . . is "still" subjected to high uncertainties. Is there a more recent reference than 2011?

Reply. Thank you for the comment. A new reference  (Glasius and Goldstein, 2016) has been added.

Line 43: Does this mean Ehn (2014)? Ehn (2012) is not listed in the references.

Reply. Thank you for the comment. We have now added the new correct reference.

Ehn, M., Kleist, E., Junninen, H., Petäjä, T., Lönn, G., Schobesberger, S., Dal Maso, M., Trimborn, A., Kul- mala, M., Worsnop, D. R., Wahner, A., Wildt, J. and Mentel, T. F.: Gas phase formation of extremely oxi- dized pinene reaction products in chamber and ambient air, Atmos. Chem. Phys., 12(11), 5113–5127, doi:10.5194/acp-12-5113-2012, 2012.

Line 58: Are there any measured O/C ratios in relevant systems that could be compared with?

Reply. Thank you for the comment. Yes Zhao et al., 2015 measured similar O/C ratios for both OH oxidation and ozonolysis of the monoterpenes with values ranging between 0.3-0.6. We have added this information in the manuscript as well.

Further they measured lower H/C ratio for SOA produced by monoterpene ozonolysis (experiments were carried out in dark with CO as OH scavenger), in comparison to OH oxidation of α-pinene and limonene, while O/C ratio were similar for both oxidation cases

Line 97: The timing seems confused: MCM+PRAM (Damian et al 2002) vs PRAM (Roldin et al 2018).

Please clarify. (Did Damian et al really refer to MCM+PRAM?)

Reply.  Thank you for the comment. I think I have made a typo by including MCM+PRAM here. It has now been removed.

Line 101: What sort of fraction of the peroxy radicals is considered in PRAM?

Reply. Thank you for the question. We have now added an explanation of the different fractions of peroxy radicals considered in PRAM. The section added is as follows:

Currently, in PRAM a maximum first generation $RO_2$ yield of 9% for α-pinene ozonolysis, 21.9 % for limonene ozonolysis, 2.5 % for α-pinene+OH, and 1% for both limonene+OH and β-pinene+OH first generation products are allowed to initiate autoxidation (Öström et al., 2017, Qi et al., 2018, Roldin et al.,

2019).

Line 130-133: Please explain whether there is likely to be any bias from using two different systems to estimate $p_0$ for different species subsets?

Reply. We have not performed any studies aimed at trying to understand the bias resulting from using 2

different systems to estimate $p_0$. We use two different $p_0$ estimation methods as the information needed (eg.

SMILES for PRAM) is not currently available to implement the same method to all compounds. Kurtén et al., (2016) have shown that Nannoolal method produces low estimates of saturation vapour pressure for multifunctional compounds due to the absence of hydro-peroxide or peroxy-acid group parameterizations.

SIMPOL on the other hand, has shown to be in better agreement with pure-liquid vapour pressures of multifunctional compounds calculated using COSMO-RS (Conductor-like Screening Model for Real

Solvents) (Eckert and Klamt, 2002; Kurtén et al., 2016).

Therefore, we have tried to use the most optimum way to utilize the current information to generate realistic
$p_0$ values.

Line 188: I think the phrase "contribution to SOA mass" is misleading. It suggests proportion of the SOA
made up by species "i", whereas the figure actually shows "SOA mass yield".

Reply. Yes I agree. I have now modified the text to:

→ In Fig. 1 the upper panel A indicates the SOA mass yields derived on applying a coupled MCM+PRAM
mechanism to ozonolysis of α-pinene and limonene (PRAM is only available for ozonolysis of α-pinene and
limonene) and the lower panel B shows ratio of yields obtained by MCM and coupled MCM+PRAM.

Lines 217 & 221, and in general: When referring to "our model" It would be helpful to distinguish at that
point which version is being used in each case (MCM+PRAM or MCM), so that the reader is reminded
whether or not PRAM is being used. The distinction is made a few sentences later: a little reorganization
would help this discussion.

Reply. In this context 'our model' refers to the MALTE-Box. We agree that its good to remind the reader
about the case being used. Hence we have now specified the version being used for each comparison.

→ Kang and Root (2007) measured a value of 0.2 for ozonolysis of α-pinene for an initial precursor VOC
concentration of 100 ppbv, while we obtain ~0.25 (MCM+PRAM) for the similar initial precursor
concentrations. The OFR yields for β-pinene (MCM-only) are significantly lower (0.02) than the values
measured by Kang and Root (2007) wherein they measured a yield of 0.49 for similar initial precursor
concentrations. Addition of seed particles promotes condensation, leading to increased SOA yields (Lambe et
al., 2015) which was confirmed by Ahlberg et al, (2019).

Line 240: The values quoted in the text for OFR-simulation SOA yield from a-pinene ozonolysis do not
match the values quoted in Table 2. Why the discrepancy?

Reply. The values in the text represented the yields for entire range of SOA mass loadings, whereas the tables
only compared yields at corresponding loadings. It has now been changed to show values for corresponding
simulated and experimental yields.

Line 270: This section needs an easier-to read introductory sentence.

Reply. This line has now been omitted. The introduction has been changed to:

→ The mass yields obtained by MCM+PRAM for α-pinene – OH oxidation are close to the measured values (Kristensen et al., 2017), while using only MCM under-predicts the mass yields (Figure 3, panel A and B, and

Table 3).

Line 277 and following: Please list in Table 3 the experimental results of Kristensen 2017 and others cited in

Section 3.2.

Reply. Done.

Line 273: It's not really true that Y=0 in these cases, since Y is the result of a comparison, and here there is nothing to compare (since there really isn't a "PRAM" for these species-oxidant combinations).

Reply. Yes true. We have omitted this line.

Line 289: It's usual to say that the model results are in good agreement with previous measurements, not the other way around.

Reply. Yes, it has been changed now.

→ For β-caryophyllene, the modeled values are in good agreement with experimental measured yields in the range of mass loadings provided by Griffin (1999) and Tasoglou and Pandis (2015).

Line 274: Please briefly remind the reader why the simulations used more surface area than the experiments?

It seems to be an important factor in the disagreement.

Reply. On re-examining, we have modified the conclusions. Changes have been made as follows:

→ Our yields for α-pinene agree well with the yields obtained by Bruns et al. (2015) where they measured yield of ~0.3 for mass loading of ~300 µg m$^{-3}$ and equivalent OH exposures. Friedman and Farmer (2018) found mass yields of 0 - 0.086 for α-pinene (ammonium sulfate seeded experiment), 0- 0.12 for β- pinene (no seed particles) and 0-0.04 for limonene (no seed particles), by varying the OH exposures between

$4.7 \times 10^{10} – 7.4 \times 10^{11}$ molecules cm$^{-3}$ s. Our simulated yields for OH oxidation of α-pinene (~0.05 - 0.31), β- pinene (~ 0 - 0.1) and limonene suggest higher mass yields for α-pinene and limonene at equivalent mass loadings, while mass yields for β-pinene are in good agreement with the experimental yields. Friedman and

Farmer (2018) suggest that the reason for this underestimation in mass yields could arise due to the exclusion of large particle sizes in the experiments and propose that these yields could represent lower bounds.

Line 302: This sentence is difficult to make sense of. Is this what is meant? "Varying NOx concentrations changes the fates of RO2 radicals formed during organic oxidation, thereby impacting . . ."

Reply. Yes your right. It has now been modified to:

Varying $NO_x$ concentrations changes the fate of $RO_2$ radical formed during organic oxidations by altering
$HO_2/RO_2$ ratio, thereby impacting the distribution of reaction products and aerosol formation (Presto et al.,
2005; Zhao et al., 2018; Sarrafzadeh et al., 2016).

Line 368: Please be specific that the PRAM contribution increases with increasing NO for NO < 1ppb. (It's
not quite clear the way it's currently written.)

Reply. Done.

Line 382: I suspect this means that the compounds shown in Fig 8 contribute >95% to the SOA mass loading
when summed in decreasing order of contribution. Please clarify the text. (It's said better in the caption and in
Line 385.)

Reply. Yes, these compounds contribute to >95% of SOA mass loadings regardless of the order of
contribution. The text has been modified to:

Figure 10 shows the most important compounds from both the MCM and PRAM that together contribute to
more than 95% of α-pinene ozonolysis SOA mass loading at 293.15 K.

Line 463: I think this means to say something like "We do not simulate appreciable mass yields from the
oxidation of BVOCS with NO3". The current text claims to describe the behavior of the actual compounds,
but I think it really intends to describe their behavior in the model. It's an important distinction.

Reply. Yes. The text has now been modified to:

The model does not simulate appreciable SOA mass yields for oxidation of BVOCs with $NO_3$, as PRAM
currently does not consider autoxidation of $RO_2$ formed from $NO_3$ oxidation of VOCs.

Comments on the Figures:

1. Why are the points in the Figures arranged in clumps/streaks? Please explain early on.

Reply. The clumps are a result of SOA mass yields for the oxidation of specific oxidant concentration with
varying BVOC concentration eg. 6 values of BVOC concentration (0 – 200 ppb) and specific oxidant
concentration ($5 \times 10^{11}$ #/cm$^3$). This has now been explained in the figure captions (Fig 1).

→ The clumps are a result of SOA mass yields for the oxidation of specific oxidant concentration with
varying BVOC concentration

2. The caption to Fig 1 says ". . . from simulations with MCM+PRAM and PRAM." Shouldn't it really say
"MCM+PRAM and MCM"?

Reply. The lower panel actually shows $Y_{(MCM+PRAM)} - Y_{(MCM)}$ or effectively the contribution of PRAM.

3. Figs 1 & 2: Please denote the two panels 'a' and b' and refer to them that way in the caption and text. This
would help the reader and might help clarify the flow of the discussion.

Reply. Done.

4. Figs 1 & 2: Please add notes to the figure captions to clarify which species are omitted from the
comparison in each case (i.e. which species use MCM-only).

Reply. Done.

5. Figs 1 & 2, 2nd panels: It is sometimes hard to figure out whether MCM makes any contribution at all.
Please make this clearer by either a) plotting the ratio $Y_{MCM} / Y_{(MCM+PRAM)}$ instead of (or in addition to) the
difference between the two, or at least b) using the same gridline interval in both panels.

Reply. Good idea. The panel B of Figures 1 and 3 now show the ratio $Y_{MCM} / Y_{(MCM+PRAM)}$

6. Please mention Figure 3 somewhere in the text, or remove it.

Reply. Done. Figure 3 has now changed to Figure 5.

Figure 5. shows the yields derived from the oxidation of BVOCs by $NO_3$. Currently, as no PRAM is available
for $NO_3$ oxidation, Figure 5 represents SOA yields derived using MCM.

Comments on the Tables:

1. Table 2: what are the figures in parentheses in the Experimental Yields column? Why are they not always
present? Please explain.

Reply. The values in the parentheses represent the corresponding mass loadings for experimental yields. We
have not added loadings for a few comparisons when the experimental loadings were similar to the simulated
loadings. We have now included all experimental loadings in parantheses.

2. Table 2: Please include the b-pinene and b-caryophyllene MCM-to-literature comparisons mentioned in the text (lines 203-212).

Reply. Done.

3. Table S1a: To make this information easier to digest, I suggest listing the compounds in descending order of contribution to SOA mass. Also: Is this just the a-pinene ozonolysis case? Please clarify. If it's for various precursors, please indicate which precursor is relevant for each product.

Reply. Done. Yes these are compounds only for ɑ-pinene ozonolysis case at different temperatures.

Minor Language Editing Suggestions:

Line 29: ". . . is still subject to . . ." (Not "subjected")

Reply. Done.

Line 134: "by contrast" might be a better phrase than "on the contrary"

Reply. Done.

Line 183 (suggestion): Move header for Section 3.1 to after "flow-tube experiments" (in line 187)

Reply. Good suggestion. Done.

Line 240: Replace "resulting" with "result".

Reply. This sentence has been omitted.

Line 243: the word "Similarly" seems strange here. (It would usually be understood to refer to the previous sentence). Perhaps this means: "As in the ozonolysis case"?

Reply. This sentence has been omitted.

Line 327-329: "Due to limited experimental constraints, PRAM presently does not consider autoxidation of

RO2 formed from NO3 oxidation of VOCS". I suggest that moving this information to the top of the paragraph would help the reader more quickly make sense of the comparisons presented.

Reply. Done.

Line 340-342 almost duplicates Line 302. It would be good to combine these two sentences, for brevity.

Reply. Done.

Line 373: ". . . the formation OF more volatile . . ." (add word "of")

Reply. Done

Line 385: ". . . and decrease (no 's') to 0.27 at 293.15K AND to 0.1 at 313.15K" (add word "and")

Reply. Done

Line 389: Maybe this is "a weak dependence . . ..WHICH becomes more pronounced. . ."? ("which", not
"but")

Reply. Done.

Line 419 duplicates some of lines 423-425. Please condense.

Reply. Done.

Line 467: (suggestion) "substantially lower than that of MCM"

Reply. Done.

Line 469: MCM *produces* more SVOCs, it doesn't just "contain" them.

Reply. Done

Line 481: delete word "respectively"

Reply. Done.

Line 477" "has paved THE way" (add word "the"). Or substitute something simpler like "helps us"

Reply. Done.

**Reviewer 2**

This is a good informative study that compares MCM and MCM+PRAM mechanisms to derive model
capabilities of known peroxy radical autooxidation mechanisms.

Thank you

Following are some comments that are recommended to improve the current work:

RC1. Add a section on experimental details before model description. This could be just a summary of various experimental studies the work is using to evaluate the model with justifications for why they were chosen.

Reply. Done. The summary is now provided in the Supplementary material.

RC2. On page 17 the authors mention T-dependence of peroxy radical autotoxidation needs further improvement/validation. More details on their assumed T-dependence in PRAM are needed. For example, what was the assumed T-dependence as a function of precursor VOC, oxidant, NO etc.? What was the T-dependence of saturation vapor pressures of SVOCs in Figure 7? Seems there are 2 different T-dependence that need to be explicitly stated: (A) T-dependence of autooxidation chemistry (B) T-dependence of their C* i.e. saturation vapor pressure or a physical process of gas-particle partitioning. This is T-dependence is a very important part and needs to be discussed clearly. Also discuss measurements of such T-dependencies as applicable.

Reply. Yes they are 2 different temperature dependence that is addressed as follows:

(A).The temperature dependence in PRAM is based on quantum chemical calculations wherein the autoxidation rates correspond to an activation energy of 24 kcal/mol. The activation energies vary for autoxidation of different $RO_2$ from α-pinene ozonolysis between 22 and 29 kcal/mol (Rissanen et al., 2015), leading to varying autoxidation rates at different temperatures (Roldin et al., 2019).

(B). The functional group contribution methods SIMPOL and Nannoolal provide temperature dependent pure liquid saturation vapour pressures. Temperature is then used as an input parameter to the calculated $p_0$.

This information has now been added to the manuscript.

RC3 Figure 7: Would it be possible to start with a VBS fit at 313 K, and then derive the VBS fit at 258K or vice versa with these T-dependencies without having to run MCM+PRAM at each of these temperatures? This is important for regional and global models that rely on VBS and cannot run full MCM+PRAM.

Reply. This could be possible but not advisable as extending VBS for varying temperatures would lead to erroneous yield estimates. VBS does not change the total number of products for varying dependencies such as $NO_x$, RH or temperatures, but rather distributes volatility of products (Donahue et al., 2009). Our analysis of different compounds contributing to mass yields at different temperatures (Figure 10, S3 and S4, Table S1-

S3) show that different products contribute to mass yields at differing temperatures. Using a VBS hence for estimating the yields derived for 258 K and extending it to 313K would result in misleading SOA mass yields.

RC4 Do the authors have any recommendations for condensed versions of MCM+PRAM that could be used in regional and global models to predict SOA yields and their oxidation state?

Reply. Yes. A condensed version of PRAM to be applied in regional and global models has been tested by reducing the number of reactions and species by lumping them into 2 sets of dimers specifically 1.representing HOM formed by ozonolysis of monoterpenes and 2. HOM formed by OH oxidation (Roldin et al., 2019). Furthermore, the author cautions that full PRAM be evaluated for conditions where a major part of $RO_2$ pool originates from precursors that do not contribute substantially to HOM formation, such as environments with high isoprene concentrations, before being applied to global and regional models (Roldin et al., 2019). More details can be found in Roldin et al., (2019). We have made this addition to the conclusions sections:

→ Furthermore, implementation of a condensed PRAM version to regional and global models has been tested but still need further validation (Roldin et al., 2019).

[revised manuscript text omitted]

**Table 1s(b)**. List of compounds contributing to > 95% of SOA mass yield at 293K. The names of compounds
are given in MCM format.

| Molecular Weight (g/mol) | Species names | Contribution (%) |
|---|---|---|
| 496 | C19H28O15 | 0.45 |
| 174.15 | C717OOH | 0.53 |
| 344 | C10H16O13 | 0.57 |
| 446 | C20H30O11 | 0.59 |
| 448 | C19H28O12 | 0.62 |
| 248 | C10H16O7 | 0.67 |
| 200.23 | HOPINONIC | 0.7 |
| 462 | C20H30O12 | 0.7 |
| 480 | C19H28O14 | 0.76 |
| 186.21 | PINIC | 0.77 |
| 188.22 | C920OOH | 0.79 |
| 510 | C20H30O15 | 0.79 |
| 325 | C10H15O11N1 | 0.8 |
| 464 | C19H28O13 | 0.82 |
| 373 | C10H15O14N1 | 0.9 |
| 178.14 | C621OOH | 1.03 |
| 478 | C20H30O13 | 1.1 |
| 246 | C10H14O7 | 1.17 |
| 341 | C10H15O12N1 | 1.2 |
| 262 | C10H14O8 | 1.26 |
| 174.19 | C811OOH | 1.35 |

| | | |
|---|---|---|
| 494 | C20H30O14 | 1.39 |
| 164.11 | C516OOH | 1.44 |
| 245.23 | C108NO3 | 1.49 |
| 162.14 | C614OOH | 1.6 |
| 220.22 | C922OOH | 1.64 |
| 200 | C10H16O4 | 1.65 |
| 204.22 | C921OOH | 1.68 |
| 357 | C10H15O13N1 | 1.93 |
| 264 | C10H16O8 | 1.97 |
| 328 | C10H16O12 | 2.2 |
| 280 | C10H16O9 | 2.35 |
| 326 | C10H14O12 | 2.41 |
| 206.19 | C813OOH | 2.64 |
| 190.19 | C812OOH | 2.73 |
| 312 | C10H16O11 | 2.77 |
| 278 | C10H14O9 | 2.86 |
| 188.22 | C97OOH | 3.03 |
| 235.19 | C813NO3 | 3.1 |
| 296 | C10H16O10 | 3.19 |
| 294 | C10H14O10 | 3.63 |
| 233.22 | C98NO3 | 3.81 |
| 261.23 | C920PAN | 4.1 |
| 247.2 | C811PAN | 4.57 |
| 310 | C10H14O11 | 6.19 |
| 216.23 | C108OOH | 6.24 |
| 204.22 | C98OOH | 7.44 |

**Table 1s(c)**. List of compounds contributing to > 95% of SOA mass yield at 313K. The names of compounds
are given in MCM format.

| Molecular Weight (g/mol) | Species names | Contribution (%) |
|---|---|---|
| 526 | C20H30O16 | 0.54 |
| 512 | C19H28O16 | 0.55 |
| 450 | C18H26O13 | 0.56 |
| 482 | C18H26O15 | 0.6 |
| 280 | C10H16O9 | 0.6 |
| 294 | C10H14O10 | 0.7 |
| 466 | C18H26O14 | 0.79 |
| 296 | C10H16O10 | 0.9 |
| 278 | C10H14O9 | 1 |
| 464 | C19H28O13 | 1.2 |
| 204.22 | C98OOH | 1.42 |
| 344 | C10H16O13 | 1.51 |
| 496 | C19H28O15 | 1.65 |
| 480 | C19H28O14 | 1.77 |
| 178.14 | C621OOH | 1.93 |
| 373 | C10H15O14N1 | 1.95 |
| 510 | C20H30O15 | 2.57 |
| 204.22 | C921OOH | 3.03 |
| 494 | C20H30O14 | 3.15 |
| 220.22 | C922OOH | 3.26 |
| 164.11 | C516OOH | 3.85 |
| 357 | C10H15O13N1 | 4.63 |
| 312 | C10H16O11 | 5.62 |
| 326 | C10H14O12 | 6.04 |

| 328 | C10H16O12 | 6.56 |
| 235.19 | C813NO3 | 6.83 |
| 190.19 | C812OOH | 6.95 |
| 206.19 | C813OOH | 7.46 |
| 310 | C10H14O11 | 18.28 |

**α-pinene - $O_3$ - CS dependence**

[Figure]

**Figure S1**. SOA mass yields for α-pinene oxidation using $O_3$ for different CS values. For the OFR runs the yields level off above a

CS value of 0.067 s⁻¹ , while chamber simulation show negligible variation with CS . Hence 0.067 s⁻¹ is selected as CS for the OFR

simulations while chamber simulations are run with 0.00067 s⁻¹.

[Figure]

**Figure S2**. Mass spectra of SOA formed from ɑ-pinene ozonolysis in the particle phase. The upper panel indicates spectra from chamber simulations while the lower panel represents the spectra from OFR simulations.

[Figure]

**Figure S3(a)**. MCM and PRAM compounds contributing to > 95% of SOA mass at 258 K and 50ppb $O_3$ and α-pinene concentrations. It can be noted that a large fraction of the PRAM species that contribute to the SOA mass at 258 K are not classified as HOM

(i.e. contain at least 6 oxygen atoms), and many of them will not be detected in the gas-phase using the present state-of-the-art

Chemical Ionization-Atmospheric Pressure Interface TOF (CI-APi-TOF) technique.

[Figure]

**Figure S3(b)**. MCM and PRAM compounds contributing to > 95% of SOA mass at 313.15 K and 50ppb $O_3$ and α-pinene concentrations.

The importance of using the MCM+PRAM scheme is illustrated in Fig. 4 which shows the relative contribution by PRAM and MCM compounds for the oxidation of α -pinene, β-pinene and limonene by OH (upper panel) and $O_3$ (lower panel) for their respective maximum SOA mass yields for both chamber and flow tube setup simulations. The present PRAM mechanism does not include the peroxy radical autooxidation products from β-pinene ozonolysis, products from oxidation of isoprene and β-caryophyllene and the products from $NO_3$ oxidation of BVOCs. Therefore, they are excluded from Fig.4.

The impact of PRAM compounds contribution to limonene ozonolysis, irrespective of chamber or flow tube setup is considered. It is evident from Fig. 9 (lower panel), which shows that upon using the standalone MCM mechanism underpredicts the SOA mass yields with PRAM compounds contributing ~ 80% and 60% respectively. For α-pinene ozonolysis, the standalone MCM scheme under-predicts the modelled mass yields by approximately 25 % and 22.5 % respectively.

[Figure]

**Figure 4.** Relative contribution of HOM and MCM compounds for selected maximum mass yields of α- pinene, β-pinene and limonene oxidation by OH (upper panel) and O₃ (lower panel) at 293.15 K.

**Summary of experimental data used for comparison**

Kristensen et al., (2017) investigated ɑ-pinene ozonolysis SOA mass yields at temperatures of 258 and 293 K. Additionally SOA mass yields from OH oxidation of ɑ-pinene were also investigated. Yields for ɑ-pinene at higher temperatures of 313 K were investigated by Pathak et al., (2007), wherein they performed experiments using ammonium sulfate seed particles. Shilling et al., (2008) performed experiments for lower concentrations of ɑ-pinene ozonolysis combinations and hence used to compare yields for loading's < 10 µg m⁻³ . Griffin et al., (1999)  used smog chambers to investigate the aerosol forming potential of various BVOCs such as β-pinene by ozonolysis and β-caryophyllene by OH oxidation. The SOA mass yields derievd from the OH oxidation of isoprene, β-caryophyllene and β-pinene were experimented by Lee et al., (2006b).

The SOA mass yields derived from the ozonolyis of α-pinene and limonene using an OFR were compared with the experimental yield from Kang and Root, (2007). The experiments also provided estimates on SOA

mass yields underestimation when performed with/without acidic seed particles in the OFR. Yields simulated from the OH oxidation of α-pinene were compared against yields measured by (Bruns et al., 2015) as they had used similar initial BVOC and oxidant concentrations. The simulated yields were also compared with ex- perimental yields from Friedman and Farmer, (2018) due to similar initial oxidant concentrations used.

---

## Editor Decision (ED1)

General comment:

- Many figures show yields vs SOA mass loading. I assume that the SOA mass loading corresponds to the resulting SOA using different concentrations of precursors (alpha-pinene etc), and not to seed aerosol loading. – Is this correct?

Could you add to the figures the information what precursor concentrations have been used to obtain the SOA mass yields? For example, I assume that the different 'branches' in Figure 6 correspond to different initial alpha-pinene concentrations (0.5 – 200 ppb)? Would the figure look the same if you plotted yield vs initial precursor concentration? – I'm not suggesting presenting all figures like this, but I'm just trying to understand it.

- Please use consistent descriptions on the y-axes (either Y or 'mass yields')

l. 65: Define MW

l. 113: Why is Table 1c cited before the other Table parts? – I think it is rather unusual anyway to have multiple Table parts (a, b, c), but I leave it up to you to or to the copy editing service to change it or not.

l. 209: The x-axis of the figure seems to only show a range of < 100 (~800?).

l. 232: replace 'value' by 'yield'

l. 252: 'ozonolysis' misspelled

l. 268: Add the value measured by Lee et al (0.58?)

l. 281: Do you mean 'Figure 3'? (Fig. 2 shows ozone oxidation.)

l. 313: Shouldn't it just read '… from application of MCM' as PRAM is not available for NO3 reactions? – If so, please also change the header of the figure.

l. 400: Shouldn't that be 99% (as in the SI, you list 98.59%)?

l. 567: Please replace by citation of ACP (not ACPD) paper.

l. 622: Reference is incomplete

l. 644: Update reference.

l. 685: Incomplete.
* * *
Supplement material

Table 1s a- c: Add to the table captions that it is for the oxidation of alpha-pinene by ozone

l. 28, 34 and 41: Should be called 'Figure 4s'

l. 36: The reference to Figure 9 seems wrong here. Do you mean Figure 4s?

---

## Author Response (AR2)

**Comments**

We thank the reviewers and the co-editor for their valuable comments. The replies are in blue.

**General comment:**

R1. Many figures show yields vs SOA mass loading. I assume that the SOA mass loading corresponds to the resulting SOA using different concentrations of precursors (alpha-pinene etc), and not to seed aerosol loading. – Is this correct?

Reply. Yes. The SOA mass loading corresponds to the SOA mass formed when the different concentrations of precursor gas is oxidized and not the seed aerosol loadings.

R2. Could you add to the figures the information what precursor concentrations have been used to obtain the SOA mass yields? For example, I assume that the different 'branches' in Figure 6 correspond to different initial alpha-pinene concentrations (0.5 – 200 ppb)? Would the figure look the same if you plotted yield vs initial precursor concentration? – I'm not suggesting presenting all figures like this, but I'm just trying to understand it.

Reply. The information of initial concentrations of precursor gas has been added to the figure 6 caption. The different branches in Figure 6 correspond to different NO concentrations which is varied between 0-5 ppb. So each branch represents mass loadings for varying $\alpha$-pinene and $O_3$ concentrations but constant NO concentrations. For example, the black diamonds correspond to mass yields from ozonolysis of different concentrations of $\alpha$-pinene (0.5-200ppb) and constant NO concentration of 0.5ppb. If the figure is plotted with initial precursor concentration on the x-axes we get the following plot. For each concentration of $\alpha$-pinene (eg 5ppb) we have 5 yield points corresponding to 5 values of $O_3$ concentrations (1.0, 5.0,10.0, 50.0,100.0 in the unit of $10^{11}$ #cm$^{-3}$).

[Figure]

Figure C1. Concentration vs Mass yields

R3. Please use consistent descriptions on the y-axes (either Y or 'mass yields')

Reply. Done. The figures descriptions on the y axes are now consistent (Figure 6 and 8).

l. 65: Define MW

Reply. Done.

l. 113: Why is Table 1c cited before the other Table parts? – I think it is rather unusual anyway to have multiple Table parts (a, b, c), but I leave it up to you to or to the copy editing service to change it or not.

Reply. Changed. The Table 1c is now cited later in a more appropriate context in the text. We follows the citing sequence of Table 1a,b and c.

l. 209: The x-axis of the figure seems to only show a range of < 100 (~800?).

Reply. Yes, That was a typo in the text.

l. 232: replace 'value' by 'yield'

Reply. Done.

l. 252: 'ozonolysis' misspelled

Reply. Corrected l. 268: Add the value measured by Lee et al (0.58?)

Reply. Done.

l. 281: Do you mean 'Figure 3'? (Fig. 2 shows ozone oxidation.)

Reply. Yes. Corrected.

l. 313: Shouldn't it just read '... from application of MCM' as PRAM is not available for $NO_3$ reactions? – If
so, please also change the header of the figure.

Reply. Done.

l. 400: Shouldn't that be 99% (as in the SI, you list 98.59%)?

Reply. Yes. Changed.

l. 567: Please replace by citation of ACP (not ACPD) paper.

Reply. Done.

l. 622: Reference is incomplete

Reply. Changed.

l. 644: Update reference.

Reply. Done.

l. 685: Incomplete.

Reply. Changed.

**Supplement material**

Table 1s a- c: Add to the table captions that it is for the oxidation of alpha-pinene by ozone

Reply. Done.

l. 28, 34 and 41: Should be called 'Figure 4s'

Reply. Yes. Changed.

l. 36: The reference to Figure 9 seems wrong here. Do you mean Figure 4s?

Reply. Yes. Changed.

[revised manuscript text omitted]